# The asymmetric Fermi surface of (Pb$_y$,Bi$_{1-y}$)$_2$Sr$_{2-x}$La$_x$CuO$_{6+\delta}$

**S. Smit** [1, *], **K. L. Shirkoohi** [2, *], S. Mukherjee [1, 3], S. Barquero [1], L. Bawden [1], E. van Heumen [1], A. P. N. Tchiomo [1], J. Henke [1], J. van Wezel [1], Y. K. Huang [1], T. Kondo [4], T. Takeuchi [5], T. K. Kim [6], C. Cacho [6], M. Zonno [7], S. Gorovikov [7], S. B. Dugdale [2], J. I. Facio [8], M. Roslova [9], L. Folkers [9], A. Isaeva [1], N. E. Hussey [2, 3, ♯], M. S. Golden [1, 10, †]

[*] *These authors contributed equally to this work.*
[†] m.s.golden@uva.nl; [♯] n.e.hussey@bristol.ac.uk

1. Van der Waals - Zeeman Institute, Institute of Physics, University of Amsterdam, Sciencepark 904, 1098 XH Amsterdam, The Netherlands
2. H. H. Wills Physics Laboratory, University of Bristol, Tyndall Avenue, Bristol BS8 1TL, United Kingdom
3. High Field Magnet Laboratory (HFML-FELIX) and Institute for Molecules and Materials, Radboud University, Toernooiveld 7, 6525 ED Nijmegen, The Netherlands
4. Institute for Solid State Physics, University of Tokyo, Kashiwa, Chiba 277-8581, Japan
5. Energy Materials Laboratory, Toyota Technological Institute 2-12-1 Hisakata Tempaku-ku, Nagoya 468-8511, Japan
6. Diamond Light Source, Harwell Campus, Didcot OX11 0DE, United Kingdom
7. Canadian Light Source, Inc., 44 Innovation Boulevard, Saskatoon, SK, S7N 2V3, Canada
8. Centro Atómico Bariloche and Instituto Balseiro, CNEA, 8400 Bariloche, Argentina
9. Institute for Solid State and Materials Research, Leibniz IFW Dresden, Helmholtzstraße 20, 01069 Dresden, Germany
10. Dutch Institute for Emergent Phenomena (DIEP), Sciencepark 904, 1098 XH Amsterdam, The Netherlands

**High-resolution angle-resolved photoemission spectroscopy (ARPES) performed on the single-layered cuprate (Pb$_{1-y}$,Bi$_y$)$_2$Sr$_{2-x}$La$_x$CuO$_{6+\delta}$ (Bi2201) reveals a 6-10% difference in the nodal $k_F$ vectors along the ΓY and ΓX directions. This asymmetry is notably larger than the 2% orthorhombic distortion in the CuO$_2$ plane lattice constants determined using X-ray crystallography from the same samples. First principles calculations indicate that crystal-field splitting of the bands lies at the root of the $k_F$ asymmetry. Concomitantly, the nodal Fermi velocities for the ΓY quadrant exceed those for ΓX by 4%. Momentum distribution curve widths for the two nodal dispersions are also anisotropic, showing identical energy dependencies, bar a scaling factor of $\sim 1.17 \pm 0.05$ between ΓY and ΓX. Consequently, the imaginary part of the self-energy is found to be 10-20% greater along ΓY than ΓX. These results emphasize the need to account for Fermi surface asymmetry in the analysis of ARPES data on Bi-based cuprate high temperature superconductors such as Bi2201. To illustrate this point, an orthorhombic tight-binding model (with twofold in-plane symmetry) was used to fit ARPES Fermi surface maps spanning all four quadrants of the Brillouin zone, and the ARPES-derived hole-doping (Luttinger count) was extracted. Comparison of the Luttinger count with one assuming four-fold in-plane symmetry strongly suggests the marked spread in previously-reported Fermi surface areas from ARPES on Bi2201 results from the differences in $k_F$ along ΓY and ΓX. Using this analysis, a new, linear relationship emerges between the hole-doping derived from ARPES ($p_{\mathrm{ARPES}}$) and that derived using the Presland ($p_{\mathrm{Presland}}$) relation such that $p_{\mathrm{ARPES}} = p_{\mathrm{Presland}} + 0.11$. The implications for this difference between the ARPES- and Presland-derived estimates for $p$ are discussed and possible future directions to elucidate the origin of this discrepancy are presented.**

## Contents

## 1  Introduction

The Fermi surface (FS) of high-$T_c$ cuprates, as deduced experimentally or via density-functional band structure calculations, is remarkably simple. Indeed, in the single-layered cuprates Bi2201 [1,2], $Tl_2Ba_2CuO_{6+\delta}$ (Tl2201) [3,4] and $La_{2-x}Sr_xCuO_4$ (LSCO) [5–7], the FS takes the form of a single warped cylinder when doped beyond the end of the pseudogap regime. Arguably, the most definitive way to determine the (Luttinger) volume of such a cylindrical FS is through quantum oscillation studies, but apart from Tl2201 [8], none of the other overdoped cuprates as yet possess sufficient crystalline purity to allow their detection.

Fortunately, the highly lamellar nature of cuprate materials facilitates the exploration of their electronic structure via $k$-sensitive techniques that probe the (near) surface region such as ARPES and scanning tunneling microscopy (STM); the former giving the most direct determination of the size and locus of the FS within the Brillouin zone (BZ). In Tl2201, the FS volumes determined by ARPES [4] and by quantum oscillations [8] show good agreement. In LSCO, where a similar comparison is not yet possible, ARPES has nonetheless determined a Luttinger volume that is in reasonable agreement with the particle density estimated from the Sr content $x$ (when $k_z$ dispersion is accounted for) [5,6,9]. In Bi2201 and $Bi_2Sr_2CaCu_2O_{8+\delta}$ (Bi2212), on the other hand, there is no direct means of estimating the hole doping $p$ due to its complex stoichiometry, the unspecified role of interstitial oxygen and the possibility of variable valence states in some of its non-Cu constituent atoms such as Bi. Hence, information on the size of the FS and its evolution across the phase diagram has been provided almost exclusively by ARPES and/or STM investigations. Bi2201 is the material of choice here, as it only has a single FS, whereas deriving hole-counts from Bi2212 requires accurate determination of both the FS's from the bonding *and* antibonding $c$-axis bilayer-split bands [10]. In ARPES studies on Bi2212, it is often only the Luttinger count of the FS associated with the bonding band that

is reported.

Precise determination of the Luttinger volume in Bi2201 by ARPES and/or STM is challenging, and large variations of order $\pm$ 0.05 holes per copper have been reported [2, 11–15] in the effective Luttinger count for a given $p$ value inferred from $T_c$ through the Presland relation [16] which we will return to later in the discussion of Fig. 5(a). Until now, the origin of this discrepancy has not been identified, though several possibilities have been proposed [15]. Reconciling these differences is of more than cosmetic value as tying together experimental data from different cuprate families and from different techniques happens via their doping level and as, ultimately, understanding the extent of the superconducting (SC) dome will be a key test of any viable microscopic theory of high-temperature superconductivity.

In this paper, we have carried out high-resolution ARPES experiments on Bi2201 single crystals spanning from the lightly underdoped to the highly overdoped, non-SC region. In the first set of experiments, high signal-to-noise energy dispersion data are acquired along the Brillouin zone diagonal, i.e. along the nodal directions. In these essentially super-structure free Pb-doped crystals, examination of the nodal data reveals a systematic inequivalence in $k_F$, with $k_F{}^{\Gamma Y}/k_F{}^{\Gamma X} \approx$ 1.06-1.10. This asymmetry echoes but exceeds the underlying orthorhombic crystal asymmetry ($a/b \approx$ 1.02) deduced by X-ray diffraction from the same crystals. From first-principle calculations, we show that this unexpectedly large $k_F$ difference can be explained by the orthorhombic crystal field, which leads to a splitting of the bands along the two nodal directions. These band structure calculations show that $v_F{}^{\Gamma Y}/v_F{}^{\Gamma X} \approx$ 1.04. Significantly, this $k_F$ enters as the 'bare band' velocity in conversion from the width of momentum distribution curves (MDCs) in ARPES to the imaginary part of the self-energy $\Sigma''$, a key descriptor of the spectral function in many perturbative descriptions of the electronic structure of interacting systems. As a result of this we can use our high-quality ARPES data to discuss $\Sigma''$ for the inequivalent $\Gamma Y$ and $\Gamma X$ nodal directions in Bi2201.

In a second set of experiments, these findings of nodal asymmetry are the catalyst for our acquisition of high-quality FS maps from these same samples, measured over the entire BZ across the same range of doping levels. In keeping with the structural data, an orthorhombic tight-binding (TB) model with two-fold symmetry in the ($k_x,k_y$)-plane is then used to model the FS, and Luttinger's theorem applied to extract $p_L$ – the number of hole-doped carriers from the FS area. Our study reveals that correctly incorporating the two-fold in-plane symmetry provides a framework in which the spread in $p$ reported in previous studies based upon an implicit four-fold symmetry can be greatly reduced. The final result is a linear relationship between estimates of $p$ from ARPES and transport-derived measurements. Similar to STM studies of the hole-doping in Bi2201 [13], we find that this linear relationship is offset by a constant, $\Delta p_o \sim$ 0.11. Although other properties derivable from both transport and ARPES such as $p^*$ are found to agree well between bulk-sensitive transport measurements and surface-sensitive ARPES (as has been shown for the same set of Bi2201 crystals [15]). The discrepancy found between the ARPES-derived total hole count $p_L$ and the transport-related carrier concentration $p_P$ is confirmed here as a robust feature that needs to be recognised, even though the reason behind it remains, as yet, unresolved.

The paper is set-up in the following way. Firstly the high-resolution nodal ARPES data for the $\Gamma Y$ and $\Gamma X$ quadrants are presented, and the differences between the two directions are quantified with regards to their MDCs widths, dispersion relations, $k_F$ values and ARPES band velocities. The impact of these measurements on the self-energies for the two directions is then discussed. The second results section involves *ab initio* theory data on the nodal dispersion relations, based on the real crystal structure of one of the crystals used for ARPES that was determined using single crystal X-ray diffraction (SCXRD: data in the SI). The third results section zooms out beyond the nodal region of $k$-space, and FS maps and their tight binding fits are presented for crystals across the doping range studied. The outcome of the resulting Lut-

tinger analysis is presented in the context of previous ARPES- and STM-based determinations of $p_L$ vs. $p_P$. The penultimate section presents a discussion of the results, also looking towards possible explanations for the differences observed between the two ways of determining the doping or carrier concentration, and the main text of the paper closes with conclusions. A substantial Supplementary Information also forms part of the paper. This covers the details of sample preparation, the analysis of the crystal structure and elemental composition of a representative Bi2201 crystal, the ARPES measurements, density functional theory calculations, and the gradient descent method used to optimise the orthorhombic tight binding fits to the experimental FS maps. The Luttinger-count analysis then follows, and finally, a comparison of additional ARPES data obtained at two different photon energies is presented.

## 2 Results

### 2.1 Nodal asymmetry

Bi2201 is known to have a distorted crystal lattice. Initially identified in the tetragonal $I4/mmm$ space group with lattice parameters $a_t = 3.8097(4)$ Å and $c_t = 24.607(3)$ Å [17], the average structure of Bi2201 was later verified to be orthorhombic (s. g. $Cccm$ [18–20] or $Ccc2$ [21–23]). There is currently no consensus whether its crystal structure is centrosymmetric or not, though several recent investigations of the related $Bi_2Sr_2CaCu_2O_{8+\delta}$ (Bi2212) system support local inversion symmetry breaking [24, 25]. Additional oxygen atoms in the BiO plane or the presence of Sr vacancies have been proposed as the drivers of the incommensurate structural modulations seen in Bi2201 [26], creating so-called diffraction replica's (DR's) that complicate the interpretation of ARPES data [27]. Fortunately, the structural modulation underpinning the DR's can be suppressed through sufficient Pb doping at the Bi site, and this was the approach taken here. At a Pb/(Bi+Pb) ratio of approximately 0.18, Bi2201 crystals free from structural modulation were successfully grown using the travelling-solvent floating-zone technique [28], and the Pb value determined using Energy-dispersive x-ray (EDX) spectroscopy from our crystals was $\sim 0.18(3)$.

Panels (a) and (b) of Fig. 1 show examples of the high-quality ARPES datasets probing the dispersion along the two nodal directions, in this case for an overdoped sample with $T_c = 23$ K (labelled hereafter as OD23K). Panels (a)[(b)] correspond to data from the XΓX [YΓY] quadrants, respectively, whereby the ΓX[ΓY] directions are aligned with the $a[b]$ axes of the orthorhombic structure. Panel (c) shows an ARPES intensity $I(k_x, k_y, E_F)$ image spanning sections of each of the pairs of ΓX and ΓY quadrants. Here the ARPES intensity maps out the Fermi surface, with the YΓY and XΓX cuts shown in blue and red, respectively. We note the absence of diffraction replica's in the FS map that would appear in panel (c) along the YΓY directions, were the suppression of the structural supermodulation found to be unsuccessful. The well-understood matrix element effect mentioned in the caption to Fig. 1 means that to measure the cut shown in panel (a) with unchanged light polarisation, the sample was rotated around the surface normal by 90$^o$ with respect to the orientation used for (c), such that the direction of the red line in the latter would then become the YΓY rather than the XΓX cut. All rotation angles of the manipulator used are reproducible to within 0.05$^o$ [30]. Multiple measurements of the nodal $2k_F$ in the two directions have been performed in immediate succession by changing only the azimuthal orientation of the sample, while keeping the exact same beamline and analyser settings. This eliminates extrinsic experimental artifacts as the source of the observed asymmetry such as stray fields or any potential influence of the absorbed photon momentum on the measured dispersions.

The momentum distribution curves (MDCs) at the Fermi energy $E_F$ for the two nodes

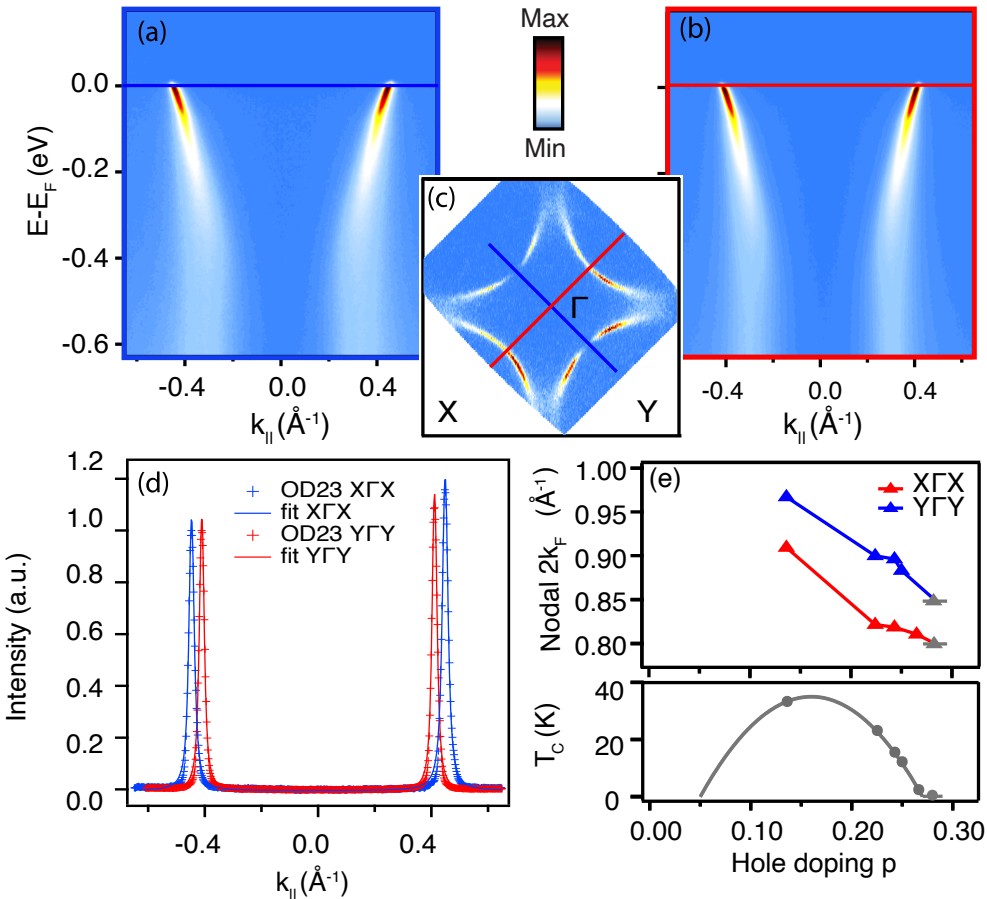

Figure 1: Nodal ARPES data measured at $T = 8$ K for an overdoped Bi2201 crystal with $T_c$ = 23 K at $h\nu = 28$ eV. (a) Nodal cut along YΓY and (b) along XΓX . (c) Full FS map used to accurately orient the nodal $k$-space cuts. Due to a known matrix element effect related to the symmetry of the initial states and the measurement geometry (horizontally polarised light with a vertical analyzer slit oriented along XΓX ), the nodal points along YΓY have zero spectral weight [29]. (d) MDCs in the two nodal directions for $E = E_F$, including two-Voigt fits. (e) Nodal $2k_F^{YΓY}$ and $2k_F^{XΓX}$ values from fits to the MDCs such as those in the previous panel, with the data now including all of the doping levels investigated. The horizontal error bar in (e) for the doping value for OD0K reflects the added uncertainty in $p$ due to an inability [15, 16] to apply the Presland formula to determine the doping level in non-SC samples.

are shown in Fig. 1(d). It is immediately evident that the dispersion relations in the ΓY and ΓX quadrants are inequivalent, despite the lack of Bi-O plane supermodulation. Given the very sharp MDC peaks, it is evident by eye that the peak-to-peak separation - equivalent to $2k_F$ - differs between ΓY and ΓX. The $2k_F$ difference can be extracted more accurately by fitting the MDCs using a Voigt line profile, which is a convolution of a Lorentzian of energy-dependent width with a Gaussian of FWHM 0.01 Å$^{-1}$ that accounts for the $k$-resolution and the imperfect flatness of the cleavage surface averaged over the photon beam. Fig. 1(e) shows the results of this fitting, collating data from all the doping levels investigated, whereby each doping level $p$ is indicated on a $T_c$ vs. $p$ sketch in the lower portion of panel (e). It is clear from Fig. 1(e) that $2k_F^{YΓY}$ is minimally 6% and maximally 10% larger than $2k_F^{XΓX}$ across the doping range studied. Seeing as this difference is evidently resolvable in high resolution ARPES data in supermodulation-free Bi2201, one wonders why this effect has not been discovered until now. One possible reason for this could be that the crystals were not physically rotated and re-

aligned to the other node when the linear polarisation was unchanged.

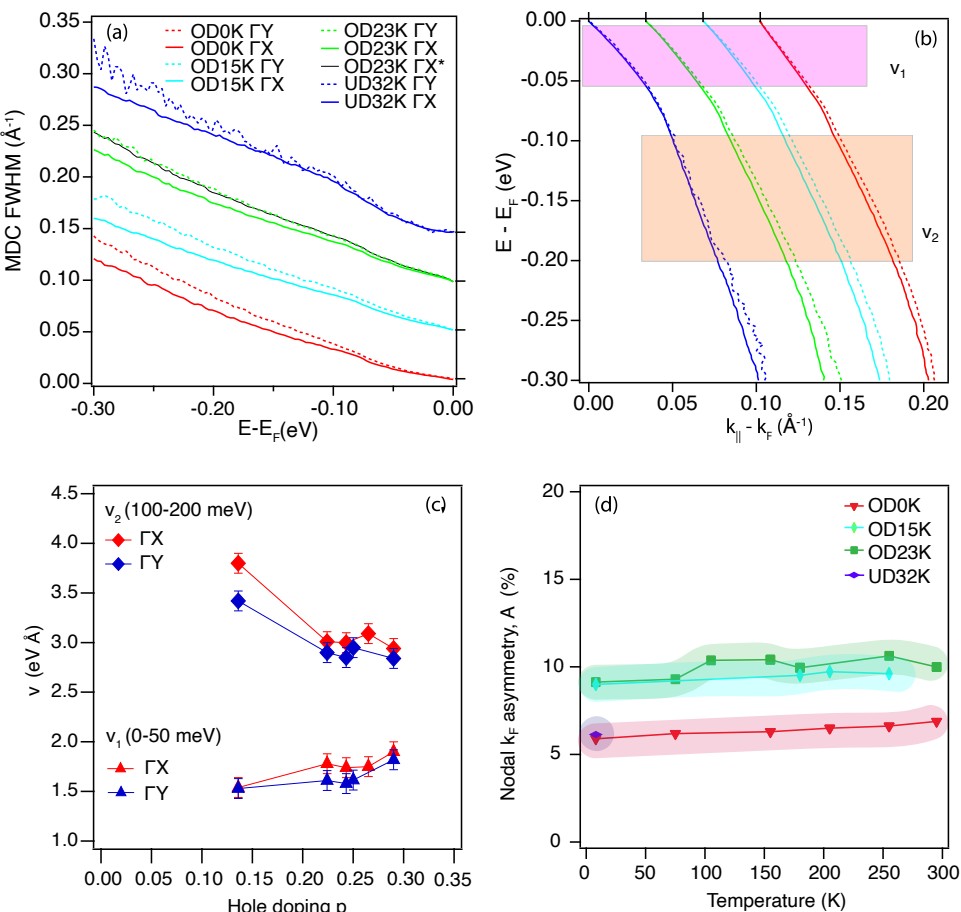

Figure 2: Analysis of nodal MDCs ($T = 8$ K) in both quadrants for the doping levels indicated. (a): MDC widths minus $\text{FWHM}_{E=E_F}$ along the ΓY (ΓX) directions shown using dashed (solid) lines, colour-coded for each doping. The traces are offset vertically for clarity. For the OD23K data, an additional curve is added (labelled ΓX*), which is the standard ΓX curve multiplied by a factor 1.13. (b): Dispersion relations determined from the loci of the MDC-fit maxima for each doping level. $k_F$ has been subtracted from the peak positions, before the traces were offset horizontally for clarity. (c): Velocity of the low-$T$ nodal dispersion extracted by a linear fit to the peak positions in (b) covering two energy regions: $v_1$ close to $E_F$ (0-50 meV) and $v_2$ at binding energies of 100-200 meV, above that of the kink visible at $E - E_F \sim 70$ meV in the dispersion curves in panel (b). (d): $T$-dependence of the nodal $2k_F$-asymmetry (defined in the main text) for each doping level. The coloured, thick shading indicates ± 1%, the level of reproducibility between data measured on different cleaves (the different quadrants for the $T$-dependence of the OD23K and OD0K data) or the same cleave (low-$T$ data for both quadrants).

Inspection of Fig. 1 instantly reveals that the $E(k_x, k_y)$ landscape possesses two-fold, rather than fourfold, symmetry. In light of this, we now investigate the relationship between the observed nodal asymmetry and the quasiparticle lifetime by fitting the width of the nodal MDCs (linked directly to the electronic self-energy) using a Voigt function. Figure 2(a) shows the width of the Lorentzian component of the Voigt fit function versus energy, along both the ΓY and ΓX directions for four doping levels. Here, we have subtracted out $\text{FWHM}_{E=E_F}$, as it is dominated by a combination of impurity scattering, $k$-resolution, and microscopic unevenness of the cleavage surface averaged over the beam spot. This enables data from different quadrants and samples / cleaves to be compared directly.

Consider for the moment the data for OD23K in Fig. 2(a) (green traces), for which an additional, thin grey trace labelled ΓX* has been added that is simply the original MDC-width data for ΓX multiplied by an energy-independent constant of 1.13. It is evident that this ΓX* curve lies right on top of the ΓY curve (green dashed line). Extending this approach for the other doping levels, it becomes clear that the MDC widths for all four doping levels differ in all cases only by an energy independent factor between 1.13-1.21. In other words, $\text{FWHM}^{\Gamma Y}$ exceeds $\text{FWHM}^{\Gamma X}$ by $\sim$ 10-20% for all doping levels spanning from UD32K to OD0K.

In the standard analysis, the nodal MDC peak widths expressed as the HWHM would be multiplied by an (assumed linear) bare band velocity, $v_{bare}$, to yield the imaginary part of the self-energy, $\Sigma''$. If, for each doping, $v_{bare}$ for the orthogonal nodal directions were identical, then $\Sigma''_{Y\Gamma Y}$ would be $\sim$ 10-20% greater than $\Sigma''_{X\Gamma X}$, inferring a 10-20% greater scattering rate for the ΓY nodes than the ΓX nodes, across the doping range studied.

In panel (b) of Fig. 2, the dispersion data measured at low temperature for the two cuts are shown for the full doping range. The measured velocity of the nodal band dispersion in each case is extracted using a linear fit to the MDC maxima in two energy regions: below the 70 meV phonon-related kink in the dispersion (between 0-50 meV indicated in the figure with pale purple shading), and above (between 100-200 meV, orange shading). The results – displayed in Fig. 2(c) – demonstrate that the ΓX node has higher velocity in both energy regions. We do not resolve $v_F$ below the very low energy kink found at ∼10 meV [31, 32], but future quadrant-resolved, high resolution laser-ARPES or low-$T$ heat transport measurements could probe its asymmetry in the very low energy range. This is of interest as this region is the most reflective of both electronic and heat transport. Bulk-sensitive heat transport studies of Bi-2212 in the so-called universal regime at very low temperatures have already shown strong in-plane anisotropy [33]. In these experiments, nodal fermions propagate heat, and the thermal transport differed by a factor of almost two between the nodal directions. This follows the identification of a deviation from the universal thermal conductivity that depends on the ratio $v_F / v_2$, with $v_F [v_2]$ being the Fermi velocity perpendicular [parallel] to the Fermi surface at the nodal points. Nodal anisotropy in thermal transport has been linked to different scattering processes (i.e. Fermi velocities) along the two nodal directions, raising the question as to which in-plane direction should be used for comparison of heat transport with ARPES [32].

The anisotropic nodal self-energies and velocities shown in Fig. 2 can also be connected to the ongoing discussion of nematicity and charge order in cuprates [34], including signatures of the existence of ordering $q$-vectors along the (tetragonal) zone diagonal [35, 36]. We note that in the iron-based superconductor LiFeAs, even anisotropy/distortion of the FS itself has been argued to reflect nematicity [37]. In the Bi based cuprates specifically, charge density modulations have been observed with resonant inelastic x-ray scattering (RIXS) to persist in underdoped and optimally doped samples, only in the Cu-O bond direction without any out-of-plane component [38]. Such translational symmetry breaking can potentially distort the FS, but will be unable to create the type of anisotropy between the two nodal directions we observe as the bandfolding only occurs along the charge order wavevector, and - if relevant - would thus affect the nodal points in the ΓX and ΓY quadrants equally. Together with our observation of clear anisotropy in samples with doping levels of up to $p \simeq 0.27$ rules out these specific ordering phenomena as the root cause. Additionally, experimental signatures of nematicity are also generally observed in the antinodal regions in $k$-space [39, 40] and typically display a marked doping and temperature dependence. Therefore, in order to explore whether the observed nodal asymmetry reported here shows a dependence on temperature or doping, we define an asymmetry parameter, A: $A = \frac{k_F^{\Gamma Y} - k_F^{\Gamma X}}{(k_F^{\Gamma Y} + k_F^{\Gamma X})/2}$. This is presented in Fig. 2(d), and the data show that at the $\pm$ 1% level, the nodal asymmetry parameter is essentially independent of temperature across the entire doping range studied.

Consequently, we find a robust and significant asymmetry in both the nodal $2k_F$ and MDC

widths that is independent of temperature and that persists across a number of important doping dependent changes: (i) the pseudogap endpoint, (ii) the end-point of the SC dome and (iii) the Lifshitz transition observed at high doping. The reliability of our data in identifying each of these key doping-dependent stages is underpinned by agreement with previous studies [2, 15, 41]. These observations therefore strongly motivate the consideration of physics *other than* nematicity, charge order or Pomeranchuk instabilities [42] to explain the origin of the nodal asymmetries described in Fig. 1 and Fig. 2.

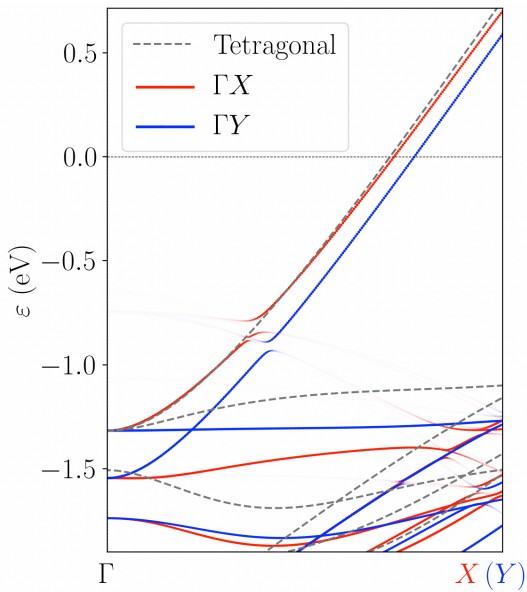

Figure 3: DFT calculated nodal band structure of (Pb,Bi)2201 using both the tetragonal (dashed black line), and the true orthorhombic (solid blue and red lines for ΓY and ΓX, respectively) structural models. The latter calculation is based on the lattice parameters from XRD given in the main text, details of which are given in the methods section (Density Functional Theory Analysis).

This conclusion sets the scene for a discussion of whether and in what way the orthorhombicity of Bi2201 plays a role in the underlying physics. Before proceeding, however, let us first discuss the degree of orthorhombicity in the crystal structure itself. To this end, single-crystal X-ray diffraction (SCXRD) and powder X-ray data were recorded at room temperature using Mo-$K_\alpha$ radiation from a piece of an OD15K crystal that was used for the ARPES experiments. The lattice parameters of the $C$-centred orthorhombic cell used to index the SCXRD pattern are $a = 5.3947(6)$ Å, $b = 24.605(3)$ Å, $c = 5.2786(6)$ Å. In the following, however, we use the convention used by rest of the cuprate community and define the $a$ and $b$ axes as the basal directions of the cuprate unit cell, and the $a/b$ ratio found to be 1.022. These and the parameters from the corresponding powder pattern are in full accordance with the previous literature [20, 23]. The atomic coordinates, anisotropic displacement parameters, and site occupancies for the Bi2201 crystal together with the details of the refinement are listed in Tables 1, 2 and 3 in the SI.

These orthorhombic structural data then formed the basis of *ab initio* calculations of the band structure, from which the most pertinent data for the nodal directions are shown in Fig. 3. The red and blue lines show the results for the orthorhombic structure, and also those from a corresponding tetragonal model as grey dashed lines, constructed by neglecting the difference in the lattice parameters $a$ and $b$ and preserving the volume per atom. Note that the bands have been unfolded from the true orthorhombic BZ, in order to enable a more straightforward comparison to the familiar tetragonal $k$-space setting. An asymmetry in the nodal $k_F$ is clearly

visible in the blue and red bands from the orthorhombic structure. The asymmetry parameter $A$ defined above can also be extracted from the DFT data and is $\sim 6.5\%$, thus at the lower end of the observed range of $A$ values obtained in our study. The DFT data enable us identify the cause of this effect which can be traced all the way down to the band bottom at the $\Gamma$ point, where bands that would be degenerate in the tetragonal setting experience an energy splitting of $\sim 200$ meV. This points to a major role of the orthorhombic crystal field acting between the (oxygen-hybridized) Cu $d_{xy}$ and $d_{x^2-y^2}$ orbitals in the origin of the $k_F$ asymmetry. Of further interest is the question of the nodal velocities, which according to DFT, could be seen as reasonable proxies for $v_{bare}$ in the context of ARPES experiments. A fit of a linear function to the data over $\pm 0.4$ eV around $E_F$ yields a Fermi velocity $v_F^{\Gamma Y}$ that is $\approx 4\%$ larger than $v_F^{\Gamma X}$, mirroring the trend between the measured velocities of the interacting bands along the two nodal directions. We also note here that the DFT-calculated nodal dispersions in both $\Gamma X$ and $\Gamma Y$ directions (shown in Fig. 3) are approximately linear over a large range in energy ($\pm$ 0.5 eV). This indicates that doping-induced changes of the chemical potential in a rigid band manner cannot explain the small variations of the asymmetry parameter we observe in Fig. 2(d), where the two extremes of our doping series (UD32K and OD0K) seem to have slightly smaller asymmetry parameters compared to the intermediate dopings. At this time, the origin of these small changes is thus an open question.

Folding in these DFT data into the analysis of the ARPES data, it is apparent that the calculated anisotropy in near-$E_F$ $v_{bare}$ values between the $\Gamma X$ and $\Gamma Y$ directions is insufficient to even out the observed opposite asymmetry in MDC widths from ARPES. It is clear, therefore, that our experimental and *ab initio* data together rule out isotropic nodal self energies in the two orthogonal directions in Bi2201. Since the data of Fig. 2 show that a simple scaling factor can be used to equalize the MDC widths over a large energy range, this means that although the absolute magnitudes of the self-energy are different, the functional form of the self-energy in both nodal directions is the same and consistent with the power-law liquid approaches presented in Refs. [43, 44], albeit with different coupling constants. The coupling constant determining the overall strength of the scattering was found to be doping independent and possibly adhering to the 'Planckian limit' of maximum possible scattering [43]. However, the results presented here suggest that the absolute magnitude of the scattering will be different in the two directions, implying that maximally one of the nodal directions could be Planckian and the other, not. Before closing this section, we note that the $k$-dependence of the power law exponents describing the self-energy in ARPES reported first in Ref. [44] - which form a key experimental argument for the possible relevance of AdS/CFT-based semi-holography calculations in accurately describing the electronic excitations of these systems - has been shown to be present for both $\Gamma Y$ and $\Gamma X$ nodal directions [45] in data from Bi2201 crystals just like those studied here.

## 2.2   Luttinger volume and hole-doping

Most analyses of ARPES and STM studies of the cuprates assume fourfold in-plane symmetry of the electronic structure coupled with a rigid band behaviour as the doping is changed, i.e. the chemical potential scales directly with $p$, while the hopping parameters remain constant. This leads to a TB parameterisation [2, 11–14] resembling Eq. (1):

$$\epsilon(k) = -2t[\cos(k_x a) + \cos(k_y a)] - 4t'\cos(k_x a)\cos(k_y a) - 2t''[\cos(2k_x a) + \cos(2k_y a)] - 4t'''[\cos(2k_x a)\cos(2k_y b) + \cos(k_x a)\cos(2k_y b)] - \mu \tag{1}$$

Here, $\epsilon(k)$ is the in-plane dispersion relationship, dependent on $k_x$ and $k_y$, $a$ is the lattice constant, $\mu$ is the chemical potential ($E_F$), and $t_0$, $t_1$ and $t_2$ are the nearest, next-nearest and next-next-nearest and next-next-next nearest neighbour hopping parameters respectively [46]. Note that the $t'''$ term is often ignored, but we include it here for direct comparison with [13].

The two-fold in-plane symmetry revealed in part I of the paper dictates that this cannot be the case in Bi2201, at least from the onset of the strange metal regime in which our study is focused. As discussed in part I, SCXRD measurements on a OD15K crystal reveal a $C$-centred orthorhombic cell with lattice parameters $a = 5.3947(6)$ Å, $b = 5.2786(6)$ Å, and $c = 24.605(3)$ Å. Thus, in order to describe the FS asymmetry, we proceed by adopting an orthorhombic TB model within the doubled unit cell and new basis vectors $a$ and $b$ rotated by $45^o$, adopting the exact lattice parameters from the SCXRD data. Furthermore, we find that the asymmetry in the nodal $2k_F$ requires that the next-nearest hopping parameters are stronger for ΓY, such that $|t_{1y}| > |t_{1x}|$. These modifications lead to the expression in Eq. (2):

$$\epsilon(k) = 2t_0\cos(k_x a/2)\cos(k_y b/2) + 2t_{1x}\cos(k_x a) + 2t_{1y}\cos(k_y b) + 2t_2\cos(k_x a)\cos(k_y b) - \mu \tag{2}$$

Here, $a = 5.39$ Å and $b = 5.28$ Å, while $t_{1x}$ and $t_{1y}$ are the (lateral) next-nearest neighbour hopping parameters in the $x$ and $y$ real-space directions, parallel to the orthorhombic $a$ and $b$ lattice vectors, respectively.

In an orthorhombic crystal where fourfold in-plane symmetry is broken but twofold symmetry preserved, the FS can be divided into two sets of identical quadrants, each pair centred along the ΓX and ΓY directions, as illustrated in Fig. 4(a). It is good to recall that even in studies based on a fourfold symmetric, tetragonal structure, the ARPES community has traditionally differentiated between the ΓX and ΓY quadrants in Bi2201 and Bi2212, due to the BiO supermodulation and corresponding DR's that arise from samples without Pb-doping. For the crystals studied here, the ARPES, SCXRD and DFT results in part I point to the need to consider each quadrant of the FS independently, irrespective of any incommensurate modulation.

Most published vacuum ultra-violet (VUV) ARPES studies on Bi2201 have been performed in the photon energy range 7 eV $\leq h\nu \leq$ 55 eV, in order to maximise the resolution of the spectra acquired, while still collecting a reasonable fraction of the BZ. Soft X-ray ARPES offers a wider field of view in $k$-space, but the signals for the Bi2201 [47] are low and, we are not aware of any FS maps or 'both ΓY and ΓX quadrant' datasets published using SX-ARPES excitation. At low photon energies, it is necessary to align the FS map so that the majority of a *single* quadrant is covered so as to collect enough data to establish a TB model. Therefore, the hole-doping concentration is typically determined following four-fold symmetrisation of the measured quadrant (implicitly assuming a tetragonal structure). Given the $k_F$ anisotropy reported in part I, this means that the $p_L$ value acquired could either be larger (i.e. fitting to the ΓX quadrant) or smaller (ΓY quadrant) than the true value of $p$, depending on how the FS nearer to the antinode behaves in each quadrant.

In Fig. 5, we present our Luttinger volume determination of the hole-doping concentration of Bi2201 across the strange metal regime at five separate doping levels. Two of these (OD12K and UD32K) were determined at $T = 8$ K, i.e. below $T_c$, while the other three (OD0K, OD3K, OD22K) were determined above $T_c$. These data were acquired at $h\nu = 100$eV (bar OD22K which was obtained at $h\nu = 113$eV), enabling at least one full (tetragonal) BZ to be visible with minimal distortion and acceptable signal-to-noise in the spectroscopic data, therefore requiring no symmetrisation. Panel (a) of Fig. 4 shows $BZ_{tetr}$, with the high symmetry points labelled, plus $BZ_{ortho}$, taking into account the difference in the in-plane lattice constants in the orthorhombic structure confirmed for these samples. The dashed dark green BZ is a copy of the black $BZ_{ortho}$, rotated by $90^o$ degrees, and the comparison of the back and green dashed BZ's helps us to appreciate the lack of fourfold symmetry in $BZ_{ortho}$. The raw ARPES data of the FS is shown for OD12K as an example, in panel (b) of Fig. 4. Shown in greyscale (dark = high intensity) is the integral of the signal over a window of $\pm 20$ meV centred around $E_F$. Panel (c) identifies the only two features seen in the ARPES data: the main FS and the so-called shadow FS. The latter, which presents as a $(\pi, \pi)$-translated copy of the main FS has been shown to be due to the orthorhombic structure of the $CuO_2$ planes, both implied by its

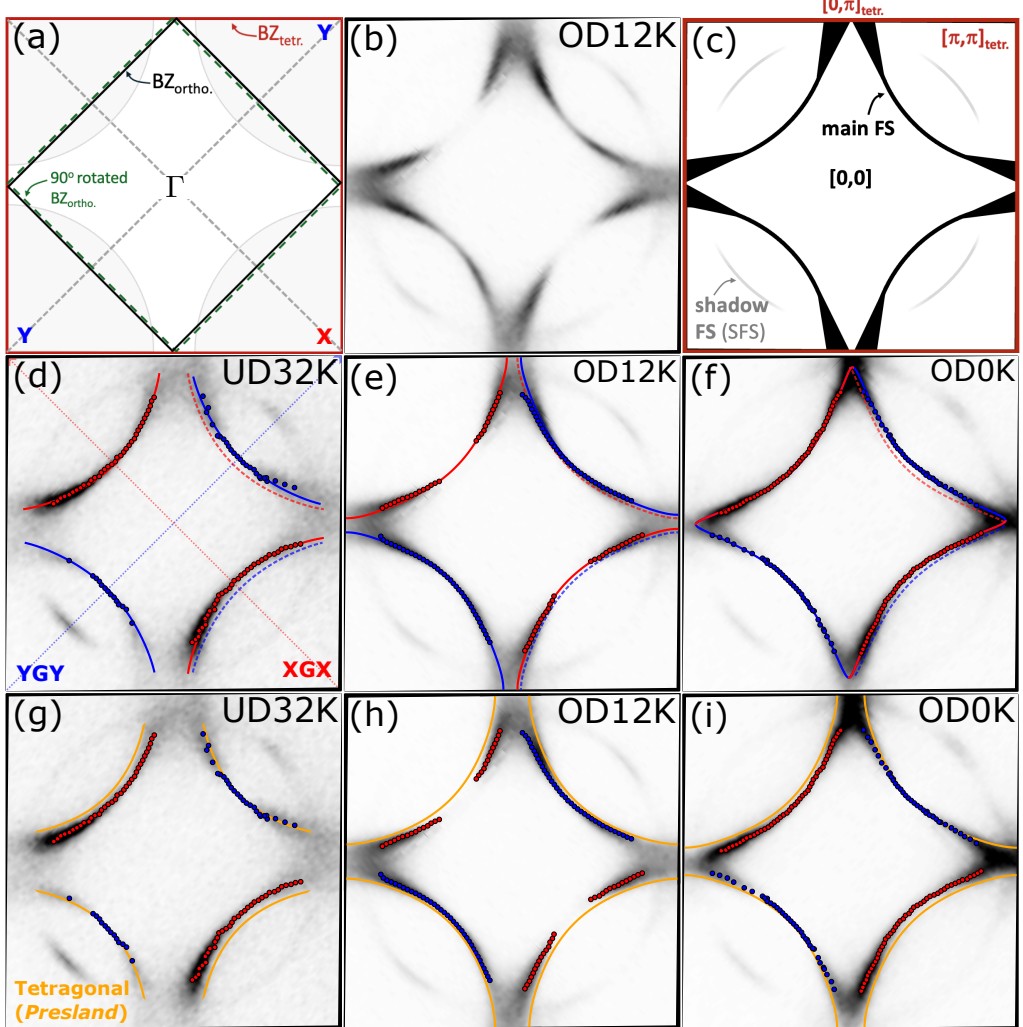

Figure 4: **Luttinger analysis of ARPES Fermi surfaces.** FS maps acquired at $h\nu = 100$eV (I05, Diamond Light Source) for three doping levels (UD32K, OD12K and OD0K). (a-c) $BZ_{ortho}$ and $BZ_{tetr}$, example raw FS dataset for OD12K and guide to the ARPES features, respectively. Panel (a) explains the $k$-space coordinates and relevant BZs, and contains a generalised cartoon of the hole-like FS in pale grey. The large (burgundy coloured) $BZ_{tetr}$ is that of the tetragonal system, and the $\Gamma$ and other high symmetry points are labelled. The solid black line shows the $BZ_{tetr}$, and the dashed green line shows the same after a 90° rotation. Panel (b) shows - as an example - the ARPES intensity at $E_F$ for the OD12K dataset, whereby the darker greyscale represents greater photoemission intensity. Panel (c) identifies the two main features that can be seen in (b): the main FS and the shadow FS. (d-i) are main data and analysis panels. MDC peak positions are superimposed on the ARPES data: red symbols for X$\Gamma$X and blue symbols for Y$\Gamma$Y. (d) UD32K. Solid, colour-coded lines show TB fits constructed by Eq. (2). The dashed-line versions show the same TB analysis, but on a 90° rotated $BZ_{ortho}$. Note that this sample is located just inside the pseudogap regime, hence Fermi arcs form inside the boundaries of the $BZ_{AFM}$, to which we have truncated the TB analysis. (e) OD12K overlaid with TB results, same colour-coding as (d). (f) OD0K overlaid with TB results, same colour-coding as (d). The Lifshitz transition means this FS is closed around the $\Gamma$ point. (g)-(i) The same ARPES, MDC fit data and colour coding as (d)-(f), now also showing as gold-coloured contours a tetragonal TB fit constructed from Eq. (1) with $\mu$ scaled such that $p_{Luttinger} = p_{Presland}$.

independence on doping and temperature [48–50], and confirmed directly by its opposing

polarization dependence in the ΓY nodal direction compared to the main FS, a result that is perfectly reproduced in one-step photoemission calculations [29].

We now turn to the Luttinger count analysis in panels Fig. 4(d-f) and (g-i). The second row of Fig. 4 shows $k_F$ loci (red symbols for XΓX and blue symbols for YΓY), determined from the centroids of the Lorentzian components of the Voigt lineshapes fitted to the $E_F$ MDCs. Also overlain on the FS data are blue [red] solid lines showing the tight binding fit using Eq. (2), based on BZ$_{ortho}$, for the YΓY [XΓX] quadrants. For all three doping levels, a second set of TB fits are also included, based on a 90°-rotated BZ$_{ortho}$. Comparing these fits to the solid TB-fit lines highlights the differences between the ΓY and ΓX quadrants. We are already familiar with this difference from Fig. 1(d) for the BZ$_{tetr}$ diagonals, but Fig. 4 (d-f) makes it clear that the asymmetry persists also away from the nodal direction. Panels (g-i) of Fig. 4 show the same experimental data and $k_F$ loci as panels (d-f), but now with a tetragonal TB model (the same type used for previous studies on Bi-based HTSCs where ΓX and ΓY symmetry is assumed) is overlain in gold. The $\mu$ in this TB model has been scaled down such that $p_L = p_P$ is shown, clearly illustrating two points: 1) if the hole-doping from the Luttinger count matches the Presland count, the TB model shows significant and systematic deviation from the measured FS, and 2) visual inspection of the ARPES FS maps confirms the need to incorporate the structural anisotropy and thus the asymmetry between the electronic states in the ΓX and ΓY quadrants to be taken into account in the TB modelling. The job now at hand is to gauge the right TB parameters for Eq. (2) (based on BZ$_{ortho}$) in an unbiased way. To help with this, a gradient descent approach (described in the SI) has been used to determine the best fit to the FS using Eq. (2) (fitting to the lowest $R^2$ value while fitting to the majority of the data i.e. avoiding local minima in favour of finding the global minimum).

Fig. 5(a) shows a compilation of Luttinger count data from the literature, plotting the correlation between $p_L$ – the doping level determined from the Luttinger analysis of FS area – and $p_p$ – that obtained from the Presland relation (to be discussed in more detail below) [16]. The solid black diagonal shows $p_L = p_P$ in both panels of Fig 5. Fig. 5(b) shows the same $p_L$ vs. $p_P$ plot for the five doping levels studied here, with $p_L$ extracted using three distinct approaches:

1. A 'true orthorhombic' fit determined using data from both the ΓX and ΓY quadrants (solid purple circles).

2. A fourfold symmetric fit determined using only data from the ΓY quadrant with larger nodal separation, $2k_F$ (blue open circles with a horizontal bar).

3. A fourfold symmetric fit determined using only data from the ΓX quadrant (red open circles with a vertical bar).

The first model accounts for the orthorhombic structure, generates an asymmetric FS, and uses $k_F$ loci from all four quadrants (i.e. the ΓX pair and the ΓY pair) to determine $p_L$ displayed as purple circles. The second and third scenarios essentially consider Bi2201 as having a tetragonal crystal structure and a FS with fourfold symmetry in the ($k_x$,$k_y$)-plane, whereby data from only a single quadrant (shown in red or blue in Fig. 4(a-f) are symmetrized to generate a full FS to determine $p_L$. It is clear that if only ΓX data are taken, the estimate of $p_L$ is too high compared to the orthorhombic model, placing the data points even further from the Presland diagonal, and vice versa were only the ΓY data to be considered. It is useful to point out that operationally speaking, fitting to the data in the antinodal region generates the largest source of ambiguity as regards the FS area and thus $p_L$. In Fig. 4, the $t_{1x}$, $t_{1y}$ and $\mu$ parameters are varied to fit to the experimental FS for the different doping levels. If we do the same for OD3K as a pertinent example (whose FS map is shown in Fig. 8), its $p_L$ is 0.3763 (see Table 4), and this is also the number that comes out when adopting the analysis using the

gradient descent approach. In Fig 8(e-f), we show how TB fits involving variation of the $t_2$ parameter can be 'imposed' on the OD3K data, with acceptably good results in the near-nodal region. From the TB contours in Fig. 8 and (f), it is evident that $t_2$ has considerable impact on the antinodal region. The Luttinger count determined by imposing a different $t_2$ choice for Fig 8(e) is 0.3513 (0.025 holes less than what we consider the best fit discussed above) and for panel (f) $p_L$ is 0.4013 (0.025 holes more than what we consider the best fit). We judge the TB fit shown in Fig 8(c) to be the best one, but we acknowledge that other ARPES-practitioners may favour those fits shown in panels (e) and (f). Consequently, in Fig. 5(b), the total uncertainty in $p_L$, expressed as the upper and lower bounds for OD3K, is 0.05 holes per copper. This clearly shows that the choice of TB fitting cannot be the source of the off-set of $\Delta p_o$ ($\sim 0.11$) in doping between $p_L$ and $p_P$ revealed in figure 5(b). The ARPES FS data for the other doping levels also lead to a similar level of ambiguity in $p_L$ (not exceeding $\pm$ 0.025), and thus in Fig. 5 this range is indicated as a vertical error bar.

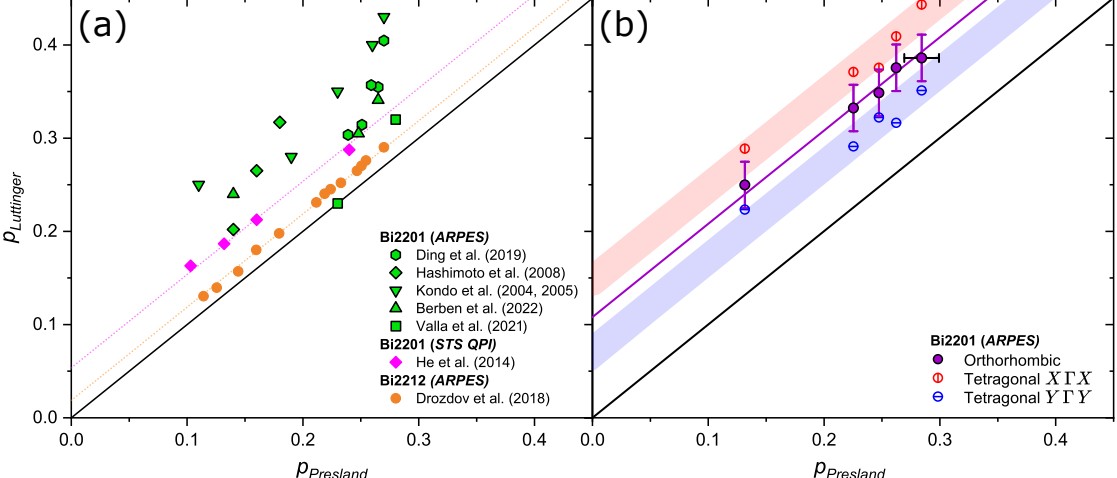

Figure 5: (a): Direct comparison of $p_L$ (the spectroscopy-derived Luttinger hole-doping) vs. $p_P$ (the $T_c$ derived Presland hole-doping) for Bi2201 and Bi2212. It is clear that $p_L$ increases with $p_P$ in all cases, but there is a scatter exceeding 0.1 in the hole-doping derived from ARPES in Bi2201. Note that STS QPI in Bi2201 clearly follows a linear trend, as well as ARPES from in situ doped Bi2212 (orange symbols). (b): ARPES-derived $p_L$ relative to $p_P$ for the range of dopings investigated here (from L to R): UD32K, OD22K, OD12K, OD3K, OD0K. The (red) $\Gamma X$ data shows the Luttinger count from symmetrizing this quadrant only, followed by fitting using gradient descent. Data in (blue) are the analogous data using only the $\Gamma Y$ quadrant. The solid black line shows $p_L = p_P$. In purple, $p_L$ is shown from an orthorhombic TB fit to FS data from both the $\Gamma X$ and $\Gamma Y$ quadrants. Note that elimination of the scatter connected to quadrant uncertainty reveals a linear correlation between $p_L$ and $p_P$, with an offset $\Delta p_o \approx 0.11$. The error bars in the absolute value of $p_L$ are $\pm 0.025$. As discussed in the text, these mainly stem from the ambiguity in fitting the FS in the antinodal region. For OD0K, a horizontal error bar is also added, representing the ambiguity about the value of $p_P$ [15], due to the absence of a SC transition (see Table 4).

In both single-layer and bilayer (hole-doped) cuprates, the SC dome is found to exhibit a near-universal parabolic dependence encapsulated by Presland *et al.* through their empirical relation ($T_c/T_c^{max} = (1 - 82.6(p_p - 0.16)^2)$) [16]. Derived originally from the Sr concentration ($x$) in LSCO, it is now broadly accepted for certain Tl-based families, Bi2212, YBa$_2$Cu$_3$O$_{7-\delta}$ (YBCO) and (tentatively) HgBa$_2$CuO$_{4+\delta}$ (Hg1201). According to the Presland formula, superconductivity extends over a doping range $0.05 \leq p \leq 0.27$ and peaks at optimal doping $p_{opt} = 0.16$. Despite the inference of universality, there are some minor exceptions [51–53]. In Bi2212, for example, the Presland dome (and hence $p_{opt}$) is shifted slightly such that $p \approx p_P$ +

0.02 [54], while in overdoped Tl2201, superconductivity is found to extend to a higher doping level $p_{SC} \approx 0.31$ [55]. In materials as complex as these, the existence of such a near-universal relation between the carrier density and $T_c$ presents a powerful constraint in our efforts to determine the pairing mechanism of high-temperature superconductivity [56].

Early on, Obertelli *et al.* also reported an empirical 'universal' relationship between $T_c/T_c^{max}$ and the room-temperature thermoelectric power $S(290K)$ [57] which was used (and later modified [58]) in conjunction with the Presland relation to estimate $p_p$. While the relation was found to hold for a number of the cuprates (including Tl2201, Bi2212, and more loosely, YBCO [59]), the universality of the Obertelli relation came into question when it was reported that it did not hold in LSCO [60, 61] nor Bi2201 [62, 63], suggesting that $p$ could not be effectively determined from thermopower studies in these materials.

Berben *et al.* later observed that across the strange metal regime, the normalised sheet resistances (i.e. resistance per $CuO_2$ layer), $R_\square(T)$, in overdoped LSCO and Bi2201 are identical for samples with the same $T_c$ [15], suggesting that $p = p_p$ in Bi2201 too. While there remains some degree of uncertainty in this method (it is not a direct measure of hole-doping), a similar coincidence [64] was also found in the magnitudes of the Hall coefficient $R_H$ in Bi2201 and Tl2201 – whose hole concentrations are known with greater confidence – thereby strengthening the notion that doping levels in Bi2201 are consistent with the Presland formula. The ARPES and STM determinations of $p$, on the other hand, present a rather different picture, as is clear from Fig. 5. Firstly, the correlation between $p_L$ and $p_P$ deduced from STM includes a constant offset $\Delta p_o \approx 0.05$ [13]. Moreover, multiple ARPES studies suggest that while $p_L > p_P$ in Bi2201, the values of $p$ measured on samples with nominally the same $T_c$ vary substantially and can be up to twice as large as those indicated by the Presland relation. Clearly, while much progress has been made in determining hole-doping using transport, ARPES and STM, reconciling results from the different techniques remains a challenge.

From this perspective, the FS asymmetry in Bi2201 revealed here appears to have resolved at least one of these issues. Inspection of panels (a) and (b) of Fig. 5 indicates that the marked variations in $p_L$ previously reported by different ARPES groups could well arise due to the specific choice of the FS quadrant from which the full FS area was then determined. In STM too, four-fold symmetrisation is typically used in order to improve data quality for FS fitting, meaning that the effect of orthorhombicity would likely be undetected in studies using this method. In the investigation of hole-doping by He *et al.* [13], the shape of the FS in UD32K Bi2201 was plotted, and comparing this to the data from our study, a strong agreement with the ARPES FS from the $\Gamma Y$ quadrant is seen. This could suggest that the values of $p_L$ from this STM study reflect only the $\Gamma Y$ quadrant, rather than the true, orthorhombic FS.

## 3    Discussion

Irrespective of the harmonisation in the estimates of $p_L$ presented here, the persistence of an offset in the $p_L$ vs. $p_P$ plot of Fig. 5(b) confirms that the size of the Bi2201 FS deduced from ARPES/STM is indeed larger than that inferred from the Hall effect or the in-plane resistivity [15]. The origin of the off-set in hole-doping, $\Delta p_o$, has yet to be identified. The off-set between $p_L$ and $p_P$ indicates that at least beyond UD32K and primarily in the overdoped regime, the superconducting dome for Bi2201 revealed by ARPES is either extended or shifted in hole-doping compared to the Presland dome. As previously highlighted, overdoped Tl2201 is also measured to possess a superconducting dome that is extended to a higher doping level, such that $p_{SC} \approx 0.31$ [55], a smaller amount than the suggested $p_{SC} \approx 0.39$ for Bi2201. The cuprates are generally unified by a universal superconducting dome, therefore a significant deviation from the nominal shape of the superconducting dome could have far-reaching im-

plications for our current understanding of superconductivity in these materials. In the case of Tl2201, the reason for the extended dome is not clear, but it is thought that the cleaner nature of Tl2201 compared to LSCO (from which the Presland dome is derived), pushes the onset of pair breaking to higher doping in the former. The persistence of superconductivity to higher doping values in ARPES-derived Luttinger hole counts in Bi2201 than seen in LSCO would seem to this challenge this pair breaking argument, as Bi2201 is regarded as the most disordered single-layer cuprate [65]. In the following, we consider a number of possible origins for the off-set, including $k_z$ dispersion, surface vs. bulk doping levels and correlation effects.

In ARPES experiments where the photon energy is varied in a systematic manner, FS slices can be extracted at different values of $k_z$. If there exists finite warping on the cylindrical FS, ARPES measurements performed at different photon energies could observe a variation in cross-sectional area and thus infer a different value of $p_{\mathrm{L}}$. Horio *et al.* applied this approach to LSCO using soft X-ray photons, resolving $k_z$ dispersion of order $\sim \pm 0.085\,\text{Å}^{-1}$ in the anti-nodal regions of the FS [6]. Zhong *et al.* further quantified this $k_z$ dispersion in LSCO in their more recent study [9]. In Tl2201, a similar but smaller dispersion ($\Delta k \sim 0.03\,\text{Å}^{-1}$, also in the off-nodal regions) has been deduced from angle-dependent magnetoresistance [3] and quantum oscillation studies [8]. To put these numbers into context, the nodal $k_F$ we find here in Bi2201 is $\sim 0.42\text{Å}^{-1}$. According to Boltzmann transport theory for a quasi-2D metal, the resistivity anisotropy $\rho_c/\rho_{ab} \propto (k_{\mathrm{F}}/\Delta k)^2$. For overdoped LSCO ($\rho_c/\rho_{ab} \sim 10^2$-$10^3$ [66]) and Tl2201 ($\rho_c/\rho_{ab} \sim 10^3$-$10^4$ [67]), this relation appears to be well respected. In Bi2201, $\rho_c/\rho_{ab} \sim 10^5$-$10^6$ [68]. Such large electronic anisotropy translates to a $\Delta k \sim 0.003\,\text{Å}^{-1}$. As detailed in the SI, our own investigation finds that $k_z$ dispersion in Bi2201 (OD23K) is nearly one order of magnitude smaller than what was measured in LSCO and lies below the effective $k$-resolution of our experiment. It is therefore very unlikely that unresolved $k_z$-dispersion could yield an offset to $\Delta p_o$ of $\sim 0.11$.

ARPES is surface sensitive, and STM even more so, and thus data from the (near) surface region is being compared to estimates of $p_L$ from bulk probes. Bi2201 and Bi2212 both possess a non-polar, low energy cleavage surface, and thus significant charge redistribution due to the formation of the cleavage surface from the original bulk crystal is not expected. As the surface terminations of Bi2212 and Bi2201 are the same, and both ARPES and STS are surface sensitive, at the straightforward level, data from both compounds and both techniques could then be expected to lie together in Fig. 5, which they do not (for in-situ doped Bi2212). Certainly, Bi2201 has a complex stoichiometry, and a key role is played by interstitial oxygen in setting the hole concentration. It is therefore possible that an as-yet undiscovered mechanism leads to $\sim$5-10% of the total holes being preferentially located at the surface, which would be enough to yield the offset $\Delta p_o$ between total carrier densities extracted from surface and bulk probes. Were the Presland formula still to hold in Bi2201 (and Bi2212), one would need to understand how this additional surface charge expands the FS but does not influence the $T_c$ value deduced in an, e.g. $T$-dependent ARPES experiment, or the value of $p^*$ from ARPES (which is in agreement between both probes).

Conversely, it could also be the case that the ARPES- or STM-derived FS volumes are in fact the correct ones, and that the $p$ values extracted from the Presland relation are being underestimated. Indeed, uncertainty in the absolute values of $\rho_{ab}$ or $R_{\mathrm{H}}$, due to uncertainties in the geometrical dimensions of the sample, is typically quoted at the level of $\sim$ 15-20%. In considering this possibility, however, it is important to make two remarks.

Firstly, in the comparative study of $\rho_{ab}(T)$ in LSCO and Bi2201 [15], it is not just the magnitude of the resistivities of samples with the same $T_c$ that were found to be identical (to within their geometrical uncertainty), but also the $T$-dependence of their derivatives $\mathrm{d}\rho_{ab}/\mathrm{d}T$ and in particular the ratio $\alpha_1(0)/\alpha_1(\infty)$ of their low-$T$ and high-$T$ $T$-linear components. In LSCO, this ratio is found to scale with $x$ in a very systematic way [69] and such a ratio is not

subject to the same geometrical uncertainties. Hence, were the carrier densities in LSCO and Bi2201 to differ by $\Delta p_o$, one would need to explain why their $\alpha_1(0)/\alpha_1(\infty)$ values coincide in samples with the same $T_c$ but not the same $p$.

Secondly, while the conversion from Hall voltage $V_H$ to Hall coefficient $R_H$ is prone to a ($\sim 10\%$) uncertainty in the determination of the thickness of these thin platelet crystals, the difference between $n_H(0)$ – the effective carrier density (per unit cell) derived from low-$T$ Hall measurements – and the carrier density derived from ARPES/STM is *far greater* than $\Delta p_o$. Indeed, one of the striking features of the cuprate strange metal is the $p$ to $1 + p$ crossover in $n_H(0)$ between $p^*$ and $p_{SC}$, corresponding to a six-fold drop in $R_H(0)$ [64]. This contrasts markedly with the ARPES result that the FS remains intact (i.e., that the Luttinger count, $n_L = 1 + p_L$) across the entire strange metal regime for $p > p^*$, (we recall there is no pseudogap here for overdoped samples with $T_c$ values of 23 K and below [15]). Finally, optical sum rule data on Bi2201 indicate a reduced carrier density $\sim 0.5(1 + p)$ across the same doping range [70].

The Hall carrier density discrepancy is arguably the most profound disconnect between the single-particle and collective (particle-particle) response within the cuprate strange metal. While the origin of this disconnect is not known at present, recent high-field studies highlighting the dual-character of the cuprate strange metal and the coexistence of quasiparticle and non-quasiparticle states [71] provide a pointer towards its possible resolution. Within this picture, only the quasiparticles contribute to the Hall response. If the two sectors are differentiated in real-space rather than in $k$-space, then ARPES may indeed detect a full Luttinger count, but only in those 'patches' that host quasiparticles. The Hall effect, by contrast, is a measure of the volume fraction of quasiparticles across the whole sample, which appears to drop monotonically as $p \to p^*$. If this scenario is correct, in the context of Bi2201, this would mean one might expect such a discrepancy to vanish at doping levels well beyond the SC dome, if then all carriers were to be coherent. The fact that STS data are collected from much smaller sample areas than ARPES, and yet both give 'above-Presland' hole counts places firm constraints on the length scale of such putative real-space patches.

An alternative scenario for this dualism is one in which the coherent and incoherent carriers are in fact momentum-space differentiated. One of the most striking features of the cuprate electronic structure is the nodal/anti-nodal dichotomy that manifests itself in the emergence of Fermi arcs (segments of disconnected FS located around the nodal regions) below $p^*$ [72] and most profoundly also in the complete lack of an antinodal quasiparticle response for $p < p^*$ in the ARPES energy spectra (EDCs) at $T > T^*$, while the spectral function at the same $k$-point shows well-defined MDC's [73] (which data from our samples also show [15]). In this $k$-space differentiated scenario, then, inside the strange metal regime ($p^* \leq p \leq p_{SC}$), Fermi arc formation is preceded by an enhanced in-plane anisotropy in the transport or single-particle lifetime around the FS as $p \to p^*$. As the example of Ref. [73] shows, the existence of MDC peaks in the normal state ARPES spectral function at $E_F$ may specify the locus of the Fermi surface, but whether the EDCs are peaked and narrow enough to indicate the presence of quasiparticles can be another matter. A future ARPES study will need to determine whether the drop in $n_H(0)$ can indeed be reconciled with the broadening of these spectral peaks around the FS. If so, then a resolution of this puzzle may finally be found.

## 4  Conclusion

In the first part of this paper, we have investigated the normal state electronic structure of modulation-free single-layer $(Pb_y,Bi_{1-y})_2Sr_{2-x}La_xCuO_{6+\delta}$ along the nodal directions of $k$-space using high-resolution ARPES. We find that even in samples without supermodulation, the nodal states are in-plane anisotropic. Across the doping range studied, the $\Gamma Y$ Fermi mo-

mentum is of order 10% greater than that along $\Gamma X$, and the Fermi velocity along $\Gamma X$ exceeds that along $\Gamma Y$. *Ab initio* calculations - based on SCXRD structural data from a representative crystal from which ARPES was recorded - capture the observed Fermi momentum asymmetry well, and point to a crystal field within the orthorhombic structure lifting the degeneracy of the bands in the two inequivalent directions. Using the DFT data to quantify the 'bare' band velocities, our data on the energy dependence of the MDC widths can be converted into a scattering rate, a process underlying innumerable papers dealing with the nodal self energy of cuprates. We find that $\Sigma''_{X\Gamma X}$ is $\sim$10 % smaller than $\Sigma''_{Y\Gamma Y}$ across the doping range studied. The lack of significant doping and temperature dependence of the observed FS asymmetry expressed in the nodal $k_F$ values suggests this effect is unrelated to the pseudogap present for $p \leq p^*$, and appears to arise directly from the orthorhombic structure of the Bi2201 system and the $CuO_2$ plane at its heart.

In the second part of this paper, we show that incorporating this asymmetry has consequences for the determination of the charge carrier concentration derived via Luttinger's theorem and is likely to be the origin of the marked scatter in previously reported ARPES-derived determination of the doping level $p_L$. Consequently, we are able to establish a new, linear relationship between $p_L$ and $p_P$ from the Presland relation, providing a robust baseline on which to discuss the disagreement between single particle (ARPES and STS) and particle-particle (transport-derived) determinations of the carrier density, and the anomalous $p$ to $1 + p$ transition seen in the strange metal phase of the cuprates. Despite the simplicity of the single band electronic structure, proximity to the (undoped) Mott insulator phase, combined with low dimensionality, still renders the electronic states at or close to $E_F$ highly anomalous. This in turn gives rise to high-temperature superconductivity and strange metallicity, both of which remain some of the most outstanding problems in modern condensed matter physics [74]. Through this work, we have identified and made more precise one of the most marked signatures of the strange metal, namely a disconnect in the carrier densities inferred from single-particle and particle-particle probes.

## Author Contributions

**Conceptualisation:** *MSG, NEH, SS, KLS*
**Methodology:** *All Authors*
**Investigation:** *All Authors*
**Visualisation:** *SS, KLS, JIF, SBD*
**Funding Acquisiton:** *NEH, MSG, EvH, JIF*
**Writing (original draft):** *SS (All + Lead on 2.1), KLS (All + Lead on 2.2), NEH, MSG*
**Writing (review and editing):** *SS, KLS, SM, SBP, JvW, T Kondo, SBD, JIF, MR, AI, NEH, MSG*
**Structure and Composition Analysis:** *MR, LF, AI*
**Density Functional Theory calculations:** *JIF (2.1), SBD (appendix, relevant to 2.2)*

## Acknowledgements

The authors would like to thank W. A. Atkinson, P. W. Phillips, M. P. Allan, T. Benschop, M. Berben, J. Ayres, J. Buhot, A. Carrington, S. M. Hayden and C. Bell for many illuminating discussions. In addition, KLS wishes to give thanks to R. D. H. Hinlopen, A. N. Petsch, M. J. Grant, M. L. Aldis and T. M. Huijbregts their valuable input and dialogue surrounding this work. Special thanks from KLS to S. Wang for insight from the perspective of STM/STS. JIF would like to acknowledge the support from the Alexander von Humboldt Foundation in the part of his work done in Germany and Ulrike Nitzsche for technical assistance.

We acknowledge the former Foundation for Fundamental Research on Matter (FOM), which is financially supported by the Netherlands Organisation for Scientific Research (NWO)

(grant no. 167METL, 'Strange Metals'). We also acknowledge the support of the European Research Council (ERC) under the European Union's Horizon 2020 research and innovation programme (Grant Agreement No. 835279-Catch-22). We acknowledge Diamond Light Source for time on Beamline I05 under Proposals SI19403 and SI22464, and Canadian Light Source for time on Beamline QMSC under Proposal 13128.

## Competing interests

The authors declare no competing interests.

## Data and code availability

The experimental datasets recorded and/or analysed, as well as the code generated during the current study are available from the authors on reasonable request.

# 5 Methods

## 5.1 Sample growth and characterisation

Near-modulation-free crystals of $(Pb_y,Bi_{1-y})_2Sr_{2-x}La_xCuO_{6+\delta}$ were grown using the travelling-solvent floating-zone technique with $y = 0.36(6)$ as determined by EDX spectroscopy. The starting chemicals (PbO, $Bi_2O_3$, $SrCO_3$ and CuO all 4N or higher purity) were well mixed. The mixture was heated in 3 to 4 temperature steps to 900C with intermediate grinding. The reacted powder was then pressed to form a cylindrical rod. After sintering it at higher temperature, the rod underwent a fast scanning in the mirror furnace. The floating-zone growth process was then carried out at a travelling rate of 0.6 mm/h under 2 atm air. Individual crystals were annealed in different atmospheres for varying lengths of time, so as to change the oxygen content and with it the hole doping controlling the carrier concentration. The critical temperatures were determined either via resistivity or AC-susceptibility measurements. The doping level $p_P$ was read-off using $T_c$ and the Presland formula [16, 62], and samples reported on here covered a doping range of UD32K from OD0K (i.e. so overdoped there is no measurable superconducting transition anymore) [44].

## 5.2 ARPES measurements

All high-resolution, nodal ARPES data reported here were recorded using (horizontally) linearly polarized light at a photon energy of 28 eV at the I05 beamline of Diamond Light Source at $T = 8$ K. The overall energy resolution was set to 12 meV, and confirmed - together with the Fermi energy position - by means of reference data from an amorphous Au film held in electrical contact with the sample. The ARPES data were measured using a commercial electron energy analyzer (SCIENTA-Omicron R4000). The full FS maps used for Luttinger counting were obtained at the Diamond Light Source end-station I05 at 100 eV (OD0K, OD3K, OD12K, UD32K) with a resolution of 12 meV at $T = 8$ K, and at the Canadian Light Source end-station QMSC (OD22K) at 113 eV at 42 meV resolution and $T = 33.8$ K. All measurements were made using linearly polarized light perpendicular to the analyser slit. The Fermi level was carefully calibrated using amorphous Au samples in electrical contact with the sample. All samples were cleaved at low temperatures, and pressures better than $1x10^{-10}$ mBar. After all temperature variations, recooling was performed and no significant aging was seen for all measurements.

### 5.3 Density Functional Theory Analysis

The Density Functional Theory (DFT) calculations shown in Fig. 3 were performed with the FPLO code [75] and based on the Generalized Gradient Approximation. For BZ integrations, we use a tetrahedron method with a mesh of $12 \times 12 \times 12$ subdivisions. The results presented are based on the experimentally-determined crystal structures detailed in SI 6.1. We have considered both a bulk calculation and a finite slab consisting of four unit cells and a vacuum of 10 Å. Both approaches yield quantitatively similar results. We have also checked in the bulk calculation that the spin-orbit coupling has a marginal effect in the orthorhombic distortion of the FS.

The Fermi surface DFT calculations displayed in Fig. 10(e) were performed using the Elk code [76]. For these calculations, structural parameters for the orthorhombic (spacegroup *Amaa*) crystal were taken from Ref. [77], and self-consistent calculations were made on 217 $k$-points in the irreducible Brillouin zone using the PBE exchange-correlation functional [78]. Hole doping of $p = 0.28$ holes per Cu was achieved by adjusting the amount of charge in the unit cell using the $chgexs$ parameter. The Fermi surface was plotted using XCrySDen [79], shown in Fig. 10.

## 6 Supplementary Information

### 6.1 Structure and Composition Analysis

Bi2201 crystals were analyzed by scanning electron microscopy (SEM, XL30 Philipps, IN400) equipped with an electron microprobe analyzer for elemental analysis using the energy dispersive mode (EDS). Elemental quantification was performed employing calibration standards for quantitative EDS and WDS micro-analysis applications. Data were averaged over 5 points of the same specimen and for several crystals of the batch. Single-crystal X-ray diffraction (SCXRD) data acquisition was accomplished on a Bruker D8 Venture (Mo-K$_{\alpha}$ radiation, $\lambda = 0.71073$ Å) diffractometer equipped with a PHOTON 100 CMOS detector. The measurement was performed at room temperature. Indexing was performed using APEX3 software. Data integration and absorption corrections were performed using the SAINT and SADABS [80] software, respectively. Crystal structure was solved by dual-space methods implemented in the SHELXT [81] program and refined by full-matrix least-squares method on $F^2$ with SHELXL [81].

Reflections in the SCXRD pattern of an OD15K crystal could be indexed as a $C$-centered orthorhombic cell with lattice parameters $a = 5.3947(6)$ Å, $b = 24.605(3)$ Å, $c = 5.2786(6)$ Å. The corresponding powder pattern (PXRD) yields a very similar set of parameters: $a = 5.3833(9)$ Å, $b = 24.5315(2)$ Å, $c = 5.2667(3)$ Å. This is in full accordance with the previous literature [20, 23]. The crystal structure was solved and refined using both $Cccm$ (No. 66) or $Ccc2$ (No. 37) space groups. The non-centrosymmetric $Ccc2$ model is supported by the $E$-statistics analysis ($|E^2 - 1|$ is 0.77, while the expected values are 0.968 and 0.736 for centrosymmetric and non-centrosymmetric space groups, respectively), as well as by the observed misbehaviour of the anisotropic temperature factors for the metal atoms in the alternative $Cccm$ structure. The atomic coordinates, temperature factors, and site occupancies for the Bi2201 crystal, together with the details of the refinement are listed in Tables 1, 2 and 3.

The crystal structure of Bi2201 possesses single sheets of corner-sharing CuO$_4$ units in which each copper atom has two additional oxygen atoms positioned above and below the sheet to form an axially elongated (Jahn-Teller-distorted) octahedron. These Cu-O planes are separated by Bi-O and Sr-O layers formed by distorted $MO_6$ edge-shared octahedra. Both the Sr and Bi bonding and geometries were found to be very irregular, e. g. Bi may form from five to six bonds to oxygen (short Bi–O distances are in a range of 2.1–2.3 Å and long ones are

about 2.6–3.3 Å) [82].

| Refined composition | $Bi_{1.902}Sr_2CuO_{5.885}$ |
|---|---|
| $M_r$ | 730.48 |
| Crystal system, space group | Orthorhombic, $Ccc2$ (No. 37) |
| Temperature (K) | 298 |
| $a, b, c$ Å | 5.3947(6), 24.605(3), 5.2786(6) |
| V Å$^3$ | 700.7(2) |
| Z | 4 |
| $\mu$ mm$^{-1}$ | 65.708 |
| Crystal size (mm) | $0.04 \times 0.03 \times 0.01$ |
| Absorption correction | Multi-scan |
| No. of measured, independent and observed $[I > 2\sigma(I)]$ reflections | 6333, 984, 808 |
| $R_{int}$ | 0.077 |
| $(\sin\theta/\lambda)_{max}$ (Å$^{-1}$) | 0.693 |
| $R[F^2 > 2\sigma(F^2)], wR(F^2), S$ | 0.0478, 0.1119, 1.089 |
| No. of reflections | 984 |
| No. of parameters | 37 |
| No. of restraints | 1 |
|  | $w = 1/[\sigma^2(F_o^2) + (0.0257P)^2 + 125.5544P]$ where $P = (F_o^2 + 2F_c^2)/3$ |
| Residual electron density $\Delta\rho_{max}, \Delta\rho_{min}$ (e Å$^{-3}$) | 3.262, -2.687 |
| Absolute structure | Refined as an inversion twin |
| Absolute structure parameter | 0.47(6) |

Table 1: Crystallographic data for (Pb,Bi)-2201 from an SCXRD experiment. Note that the Pb content is replaced by Bi because it is impossible to refine both elements simultaneously by X-ray diffraction.

| Atom | $x/a$ | $y/b$ | $c/z$ | $U_{eq}$ | Occupancy |
|---|---|---|---|---|---|
| Bi | 0.2676(1) | 0.43697(4) | 0.27412(2) | 17.8(3) | 0.95(1) |
| Sr | 0.2466(4) | 0.6774(1) | 0.2540(8) | 19.4(9) | 1 |
| Cu | 1/4 | 3/4 | 0.757(1) | 16(1) | 1 |
| O1 | -0.003(8) | 0.7517(8) | 0.513(6) | 20(4) | 1 |
| O2 | 0.231(3) | 0.3544(7) | 0.247(5) | 20(4) | 0.94(5) |
| O3 | 0.650(4) | 0.4322(9) | 0.374(4) | 20(4) | 1 |

Table 2: Atomic coordinates and isotropic displacement parameters (Å$^2$ $\times 10^3$) for (Pb,Bi)-2201. $U_{eq}$ is defined as 1/3 of the trace of the orthogonalised $U_{ij}$ tensor. Note that the Pb content is replaced by Bi because it is impossible to refine both elements simultaneously by X-ray diffraction.

Generally, our SCXRD data confirm the average crystal structure of Bi2201 as described in the literature. In distinction to [20], where the Pb dopant was distributed across both Bi and Sr sites, in our Bi2201 crystal the Sr sites are occupied predominantly by Sr, whereas Pb/Bi sites have mixed occupation. Since the scattering powers of Pb and Bi are almost equal for wavelengths which are not close to an absorption edge, these atoms cannot be distinguished

| Atom | $U_{11}$ | $U_{22}$ | $U_{33}$ | $U_{23}$ | $U_{13}$ | $U_{12}$ |
|------|----------|----------|----------|----------|----------|----------|
| Bi | 15.2(5) | 22.7(5) | 15.5(5) | 4.4(6) | -4.1(12) | 0.5(4) |
| Sr | 12.1(11) | 29.3(13) | 16.7(19) | -0.4(16) | -5(2) | -2.3(8) |
| Cu | 9.2(18) | 28(2) | 11(3) | 0 | 0 | 0.5(15) |

Table 3: Anisotropic displacement parameters ($\text{Å}^2 \times 10^3$) for the metal atoms in (Pb,Bi)-2201. The anisotropic displacement factor exponent takes the form: $-2\pi^2[h^2a^{*2}U_{11} + 2hka^*b^*U_{12} + ...]$.

by conventional X-rays. The experimental composition measured by EDS and averaged over all measurements is found to be Pb:Bi:Sr:Cu = 0.35(3):1.69(8):2.01(9):1. The Pb/(Bi+Pb) ratio is 0.175. Our SCXRD refinement reveals that the Pb/Bi site may be slightly under-occupied at 95(1) % (see Table 2), as is also independently confirmed by the EDS data.

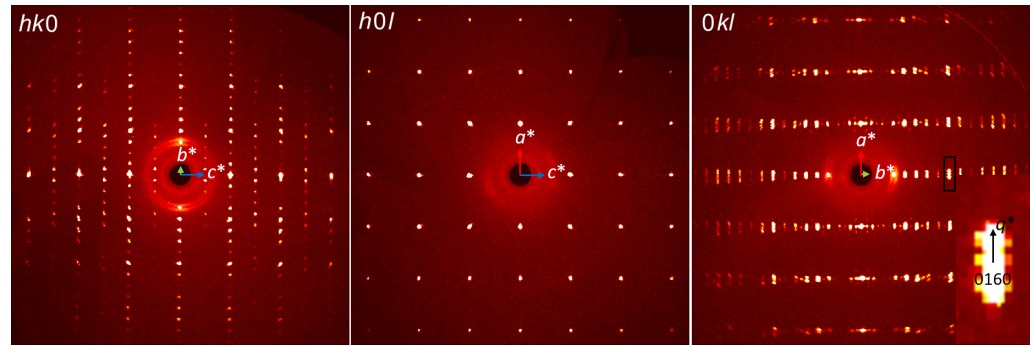

Figure 6: Recalculated reciprocal layers $hk0$, $h0l$ and $0kl$ for the one (Pb,Bi)-2201 crystal that did exhibit weak incommensurate modulation with $q \approx 0.2c*$. Those used for ARPES did not show significant signs of the incommensurate modulation. [82]

While this degree of Pb doping is known to effectively suppress the incommensurate modulation for ARPES experiments on Bi2201, we still find its weak presence in the crystals under investigation. In most crystals, this modulation is rather poorly pronounced and subject to disorder, but we were able to find one crystal, whose reciprocal space reconstructions clearly show a weak incommensurate modulation with $q \approx 0.2c*$, e.g., in the ($0kl$) reciprocal layer (Fig. 6). This finding again accords well with the observations by [23] and [83] that have shown that $q_1 \approx 0.23a^*$ satellites may still occur for Bi-rich (Pb-poor) doped specimens and they can be additionally accompanied by $q_2 \approx 0.14a^*$ satellites for Bi-poor crystals.

## 6.2 Hole-doping and Luttinger's Theorem

In order to calculate the number of hole-doped carriers, a TB model is often first guessed and then fitted by trial-and-error methods, which could limit the accuracy of the results. In the case of $n = 2$ nearest neighbours, this is reasonable, but as $n \to 3$ or 4, the number of variables that can be adjusted becomes significantly more challenging and cannot be done within a reasonable time-frame by hand, and often a rigid-band model is assumed, i.e. only chemical potential changes, where this is often not the case, as we show in this paper. Here, we propose a straightforward computational method using an Python program for determining the optimal TB model and parameters to describe the two-dimensional band structure obtained from ARPES experiments. Using a machine learning method known as gradient descent (GD)

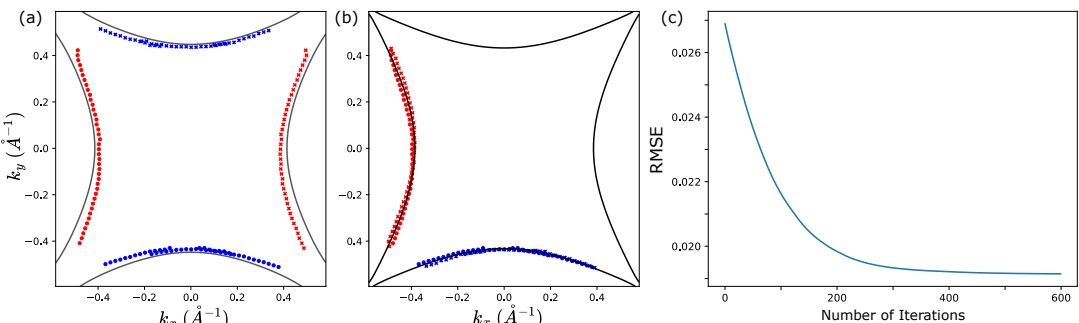

Figure 7: Gradient Descent applied to fitting the TB model of the OD3K Bi2201 FS (see Figure 8 panels (b) and (d)). (a): Initial guess TB model. The initial parameters are the same as OD0K in Table 4, but with $\mu = 0.34$. (b): The final, converged, best-fit model (the final parameters can be found in Table 4), with data from opposing quadrants, as well as the shadow band (in the case of nodal extinction present in the $\Gamma Y$ quadrant here due to the polarisation of light) is transformed onto a single quadrant to aid convergence, and reduce the effect of residual analyser distortion. The convergence parameters are found in Table 4 (OD3K). (c): Plot of root mean-squared error (RMSE) against number of iterations (commonly referred to as epochs). Gradient Descent process minimising the cost function between the MDC peaks and the TB model $k_x$ and $k_y$ data is minimised to convergence at 1000 iterations for $\mu$.

where the bias (chemical potential) and weights (hopping parameters) are optimised to minimise a cost function over many epochs, this converges at the best-fit TB model describing the band structure with significantly improved fitting accuracy ($R^2$ value). Care must be taken, however, to give a reasonable initial guess - it is possible to find a local minima of the cost function (which describes how well the current fit describes the available data set) rather than the global minimum, particularly when fitting to data that are too noisy. In these cases, stochastic GD can be used, but the implementation of this is beyond the scope of this work [84].

An initial guess for the TB model chemical potential (bias) and hopping parameters (or weights), $\mu, t_0, t_{1x}, t_{1y}$ and $t_2$, is input into the algorithm and a learning rate, $\alpha$, is defined for parameters $\alpha_\mu$, $\alpha_{t_1 x}$ and $\alpha_{t_1 y}$. TB models are often taken as 'rigid', meaning that a single parameter (usually $\mu$) scales with doping and the rest are kept fixed. We know from part I that measured crystal orthorhombicity requires that $t_{1x} \neq t_{1y}$ and that these parameters are changing such that the nodal $2k_F$ separation varies between 6-10% across the doping range studied. We keep both $t_0$ and $t_2$ fixed, and apply GD to an initial set of data (in this case, OD0K) to model the whole FS and converge on a best Ansatz for the rest of the data, which fits to data from both the $\Gamma X$ and $\Gamma Y$ region simultaneously. We use this Ansatz, and carry out the following steps to accurately reproduce a TB model for each of the FS maps from the other doping levels: (1) Set $\alpha_\mu$ small but non-zero, and large enough to converge within 1000 iterations (i.e. 0.005) with $\alpha_{t_1 x}$, $\alpha_{t_1 y} = 0$. (2) Repeat for $\alpha_{t_1 x}$, with $\alpha_\mu, \alpha_{t_1 y} = 0$. (3) Repeat for $\alpha_{t_1 y}$, with $\alpha_\mu, \alpha_{t_1 x} = 0$. (4) Finally, repeat for $\alpha_\mu$, with $\alpha_{t_1 x}, \alpha_{t_1 y} = 0$. These steps should be taken separately for each parameter as above so that the ideal convergence can be reached without parameters simultaneously 'cancelling out' thereby artificially reducing cost. Our machine learning method considers the position of every MDC peak and its distance from the initial guess defined by root mean squared error (RMSE) and using GD, the RMSE is minimised for a number of steps (usually of the order 200-1000) until convergence. In order to increase the accuracy of the fitting procedure, MDC peaks from the main band and shadow band are translated by unit-vectors in $k$-space onto a single appropriate $\Gamma X$ or $\Gamma Y$ quadrant (this process is shown progressively from Fig. 7(a-b), with a single-step convergence shown in Fig. 7(c)), and GD is performed on these regions. We define the equation in terms of weights

($w$), bias ($\mu$), and a learning rate ($\alpha$). The weights and bias are adjusted relative to their individual learning rate. For each step $i$ in an 'epoch' (the total amount of steps), the weights are updated as follows (and the bias similarly, but in 1D):

$$w_{i+1} = \begin{matrix} t_{0,i+1} \\ t_{1,i+1} \\ t_{2,i+1} \end{matrix} = w_i - \Sigma_{i=1}^{n} \frac{d(MSE)}{dw} \tag{3}$$

Convergence occurs when the RMSE between the contour and MDC peaks (cost) of the TB fit is at its smallest. The GD algorithm [85] will then converge when the minimum of the cost function has been reached. An initial guess also avoids the pitfall of converging on a local minimum, rather than a global minimum (which is desired), but this is not essential in our case. At convergence, the GD program (written in python) will generate the 'best fit' parameters $\mu, t_0$ (rigid), $t_{1x}, t_{1y}$ and $t_2$ (rigid). See Table 4 for the orthorhombic TB parameters obtained from our study: the results for UD32K, OD12K and OD0K are shown in Fig. 4 and the results for the two remaining dopings are shown in Fig. 8. The relationship between doping and these parameters is shown in Fig. 9, where the chemical potential $\mu$ scales with doping, but there is no clear relationship with $t_{1x}$ or $t_{1y}$.

| Sample | $p_P$ | $p_L$ | $\mu$ | $t_0$ | $t_{1x}$ | $t_{1y}$ | $t_2$ |
|---|---|---|---|---|---|---|---|
| OD0K (I05) | 0.27-0.30 | 0.3866 | 0.3762 | 0.49046 | -0.0896 | -0.0999 | -0.00154 |
| OD3K (I05) | 0.263 | 0.3763 | 0.3750 | 0.49046 | -0.0855 | -0.1039 | -0.00154 |
| OD12K (I05) | 0.248 | 0.3491 | 0.3687 | 0.49046 | -0.0919 | -0.0979 | -0.00154 |
| OD22K (CLS) | 0.226 | 0.3328 | 0.3605 | 0.49046 | -0.0883 | -0.0983 | -0.00154 |
| UD32K (I05) | 0.132 | 0.2501 | 0.3164 | 0.49046 | -0.0801 | -0.0988 | -0.00154 |

Table 4: Orthorhombic TB parameters (following equation 2) for the dopings studied in this paper.

| Sample | $p_L \approx p_P$ | $\mu$ | $t$ | $t'$ | $t''$ | $t'''$ |
|---|---|---|---|---|---|---|
| OD0K (I05) | 0.285 | 0.274 | 0.216 | -0.041 | 0.036 | -0.015 |
| OD3K (I05) | 0.263 | 0.2625 | 0.216 | -0.041 | 0.036 | -0.015 |
| OD12K (I05) | 0.248 | 0.2547 | 0.216 | -0.041 | 0.036 | -0.015 |
| OD22K (CLS) | 0.226 | 0.2427 | 0.216 | -0.041 | 0.036 | -0.015 |
| UD32K (I05) | 0.132 | 0.185 | 0.216 | -0.041 | 0.036 | -0.015 |

Table 5: Tetragonal TB parameters (following equation 1) for the dopings studied in this paper.

At the Fermi energy, Luttinger's theorem [86, 87] can be used to determine the carrier density of Bi2201. Assuming negligible $c-$axis dispersion (motivated by the strong in-plane v.s. out-of-plane resistivity anisotropy), this involves calculating the ratio of the hole-doped area to the electron-doped area within a nominal BZ. In the case of an orthorhombic crystal, this becomes slightly complicated in the first BZ due to band-folding, so it is easier to apply Luttinger's theorem to sum of the 1st and 2nd BZ's (which together are almost the same as the first tetragonal BZ, but in the shape of a parallelogram rather than a rectangle). The FSs for both the nearly-tetragonal and orthorhombic doubled-area unit-cell are shown in panels (a) and (b) of Fig. 10, and their respective unit-cells in panels (c) and (d). Panel (e) shows the results of DFT (Fermi surface calculations detailed in the latter part of 5.3) using the orthorhombic crystal structure and lattice parameters from the unit-cell in (d). In the case of the 'tetragonal sized' unit-cell and larger BZ shown in Fig. 10(a), Luttinger counting simply follows the $n = 1 + p$ rule:

$$p_{Luttinger} = 2\frac{A_{blue} + A_{red}}{A_{BZ}} - 1 \tag{4}$$

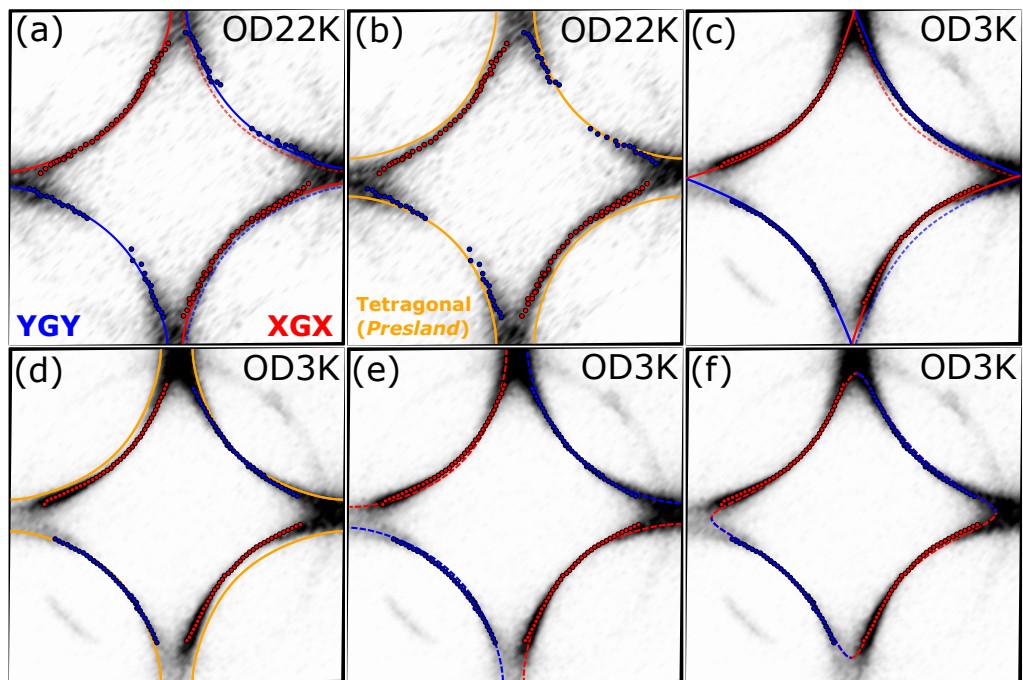

Figure 8: $k$-space FS maps acquired at $h\nu = 100$eV (I05, Diamond Light Source), as described in Figure 4. (a) OD22K: twofold in-plane symmetric (blue ΓY, red ΓX) and (b): fourfold in-plane symmetric (gold) TB fits corresponding to $p_{Luttinger} = p_{Presland}$ respectively. (c) OD3K: $C_2$ (blue ΓY, red ΓX) and (d): fourfold in-plane symmetric (yellow) Presland count TB fits. (e-f): Two-fold in-plane symmetric TB fits as in (c), but employing variation in the $t_2$ parameter to illustrate the ambiguity in the Luttinger count connected with the fits to the antinodal regions of the FS. For (e), $t_2 = 0.007$, $p_L = 0.352$. In (f), $t_2 = -0.010$ and $p_L = 0.400$. Consequently, for these OD3K data, the difference in $p_L$ between these two extreme fits is $\sim 0.05$ holes per copper

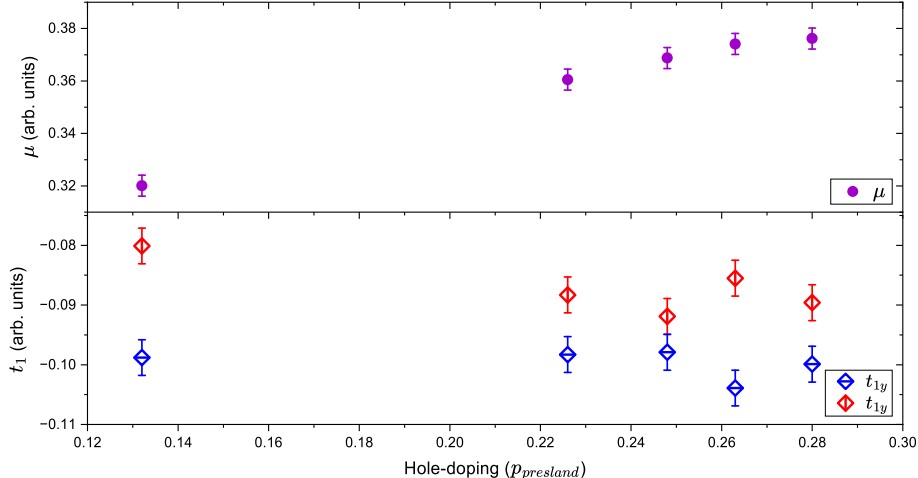

Figure 9: Variable parameters from the quasi-rigid TB model for each doping, as laid out in Table 4 shown. Note that $\mu$ is generally increasing with doping, but $t_{1x}$ and $t_{1y}$ are varying, indicating some more or less rend-less variation in orthorhombicity. Dashed line showing each different sample, from left to right: UD32K, OD22K, OD12K, OD3K and OD0K. Error bars $\mu : \pm 0.004$, $t_{1x}$, $t_{1y} : \pm 0.003$

However, in the case of the, smaller, orthorhombic BZ, an alternate expression is required equivalent to Eq. (4) where band folding must be accounted for (except in the gapped regime,

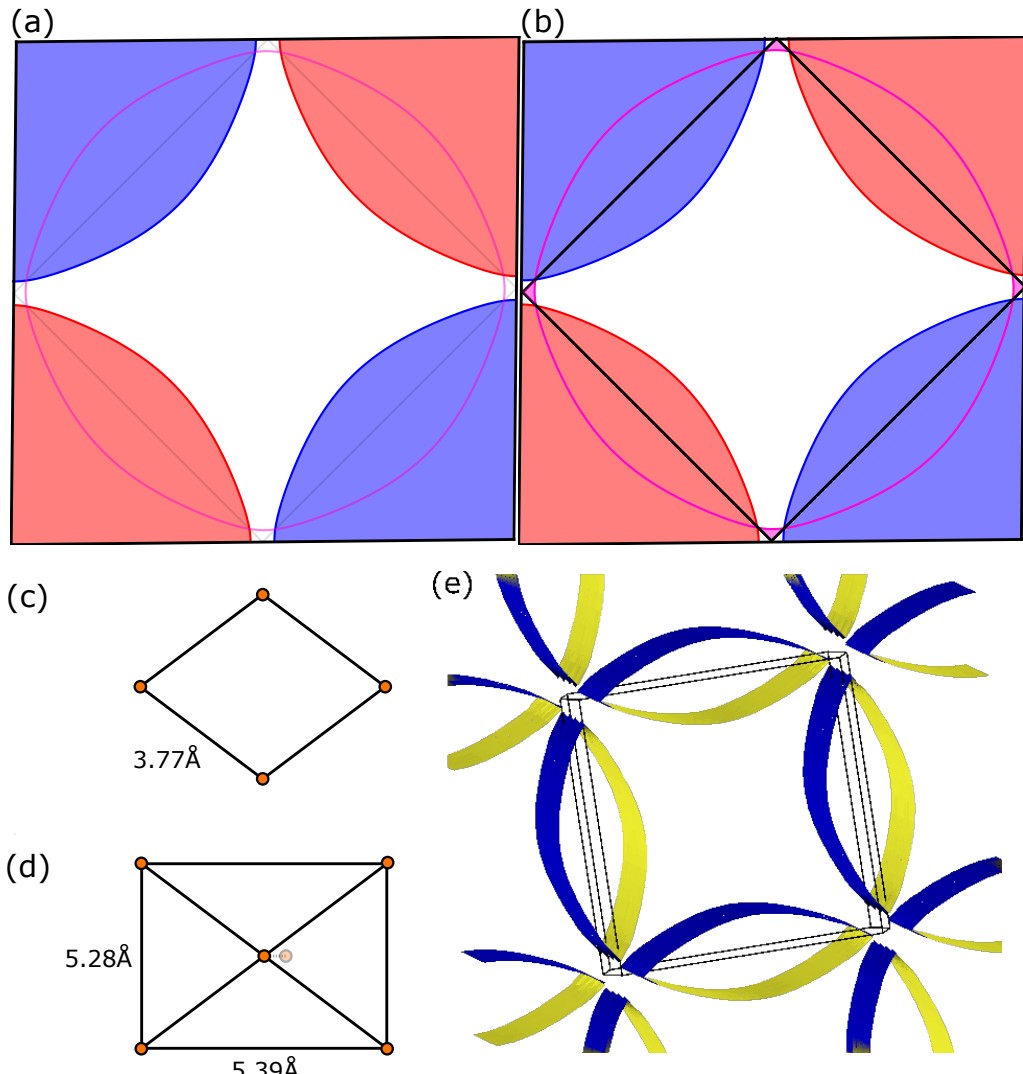

Figure 10: Luttinger counting, Fermi surface (FS) and unit-cell geometry used in this study. (a): The TB model for the FS of the OD22K data (see Figure 8 (a) for the full data). The regions in red and blue are added together in Eq. (4) to calculate $p$. (b): The same FS as in (a), but in the orthorhombic regime including the band-folding represented in pink following Eq. (5). (c): The 'tetragonal' real-space, primitive unit-cell for Bi2201. (d): The 'orthorhombic' unit-cell (aspect rato greatly exaggerated), with an off-centred Copper atom to illustrate one of the suggested reasons for the origin of the shadow band. (e): LDA-DFT calculations within the orthorhombic unit-cell regime showing the approximate FS of OD22K (blue and yellow contours), where one can indeed see the orthorhombic BZ (in white) matches (b).

such as pseudogap in UD32K where an $n = p$ FS is suggested by STS [13]):

$$p_{Luttinger} = \frac{A_{blue} + A_{red} - A_{pink}}{A_{BZ}} \tag{5}$$

The shadow band, which is used to calculate the pink area in Eq. (5) and is illustrated in Fig. 10(a,b) in pink, and has been well studied and does not show a temperature or doping dependence (other than a $p$−dependent area in change like that of the main FS), and originates from the orthorhombic distortion in crystal structure [29, 48–50]. Furthermore, the argument that the shadow band originates from the orthorhombicity could be justified by an off-centre

copper atom, as illustrated in Fig. 10(d), which would nullify the use of the tetragonal scheme, even if $a = b$. It should be noted that to our knowledge, in either regime, these two equations for hole-doping concentration are equivalent in the strange metal regime such that $n = 1 + p$.

The problem of hole-doping inconsistency should also be considered through the lens of the 3D Fermi surface. Application of the Luttinger sum rule to determine the number of hole-doped carriers on the 2D FS only holds in the case that there is no $c-$axis (or $k_z$) warping present in the system. In Bi2201, previous work by Kondo *et al.* [11,88] suggests that anti-nodal dispersion has doping dependence such that in the strongly overdoped region (OD7K, OD0K) there is $k_z$ dispersion of around 10 meV at most, but is negligible approaching optimal doping (OD22K and OP35K). Even at 10 meV, the dispersion measured by the Kondo group is below the resolution of the FS maps used to calculate hole-doping and are within the error of our experiment. Horio *et al.* [6] discovered significant dispersion (larger than Bi2201) in the anti-nodal regions of the significantly less two-dimensional La-based cuprates (LSCO and Eu-LSCO), by probing at different soft X-ray photon energies. The variation in dispersion varies smoothly with a period matching the $k_z$ extent of the 3D BZ,, revealing the intrinsic 3D structure of the material. Zhong *et al.* further quantified this dispersion relation in their recent study of LSCO [9]. In Tl2201, a similar but much less significant $c-$axis warping effect has been measured using both ARPES and quantum oscillation studies in the off-nodal regions of the FS [4,55,89]. These kind of effects can result in a different value of $p_L$ being measured by ARPES at different photon energies ($h\nu$). There are many factors involved in the choice of $h\nu$, and in practice it is a trade-off between resolution (favours lower $h\nu$), $k$-space region collected (favours higher $h\nu$) and photo-emission intensity which is certainly reduced for soft x-ray energies [90].

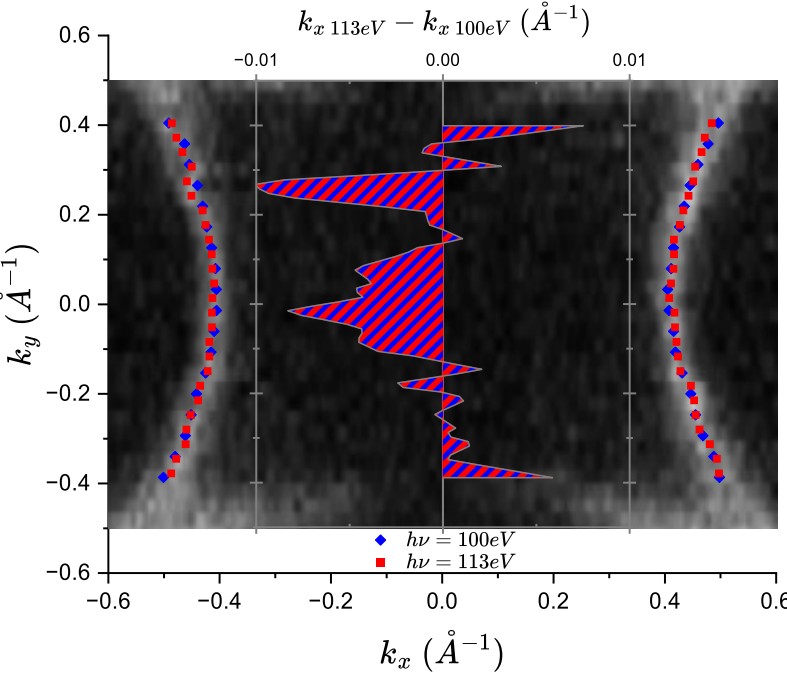

Figure 11: Bi2201 OD22K 113eV Fermi surface (FS) map measured with $h\nu = 113eV$, overlaid with MDC peaks from $h\nu = 100eV$ data (blue) and from the $h\nu = 113eV$ data (red). In the center of the image, the difference between the position in $k_x$ of the two FS is plotted using the upper-most $k_x-$scale in grey, with filled red and blue lines show difference between the two contours in $\mathring{A}^{-1}$. The lower-most (black) $k_x$ scale shows the dimensions of the FS map, while the $k_y$ axis is shared between both the FS map and the contour difference plot. It is clear that the difference in $k_x$ is less than the resolution of the experiment.

At the Canadian Light Source, the FS of OD22K Bi2201 was measured at two different photon energies ($h\nu = 100eV$ and $113eV$), for which the $k_z$ values differ by close to $\frac{\pi}{c}$, i.e. half of the $k_z$ dimension of the 3D BZ. By fitting MDC peaks to the FS at each photon energy between the node and up to $30^o$ off-nodally, and quantitatively comparing the difference, the position of the two FS contours were found to show negligible difference ($\sim 0.01\text{Å}^{-1}$ at most) as shown in Fig. 11, far below the resolution of the FS map. This result is in agreement with previous findings of the Kondo group, suggesting that periodic $c-$axis dispersion is not the cause of the apparent enhanced carrier density seen in ARPES compared to the Presland relation figures in Bi2201.

## 6.3 MDC analysis

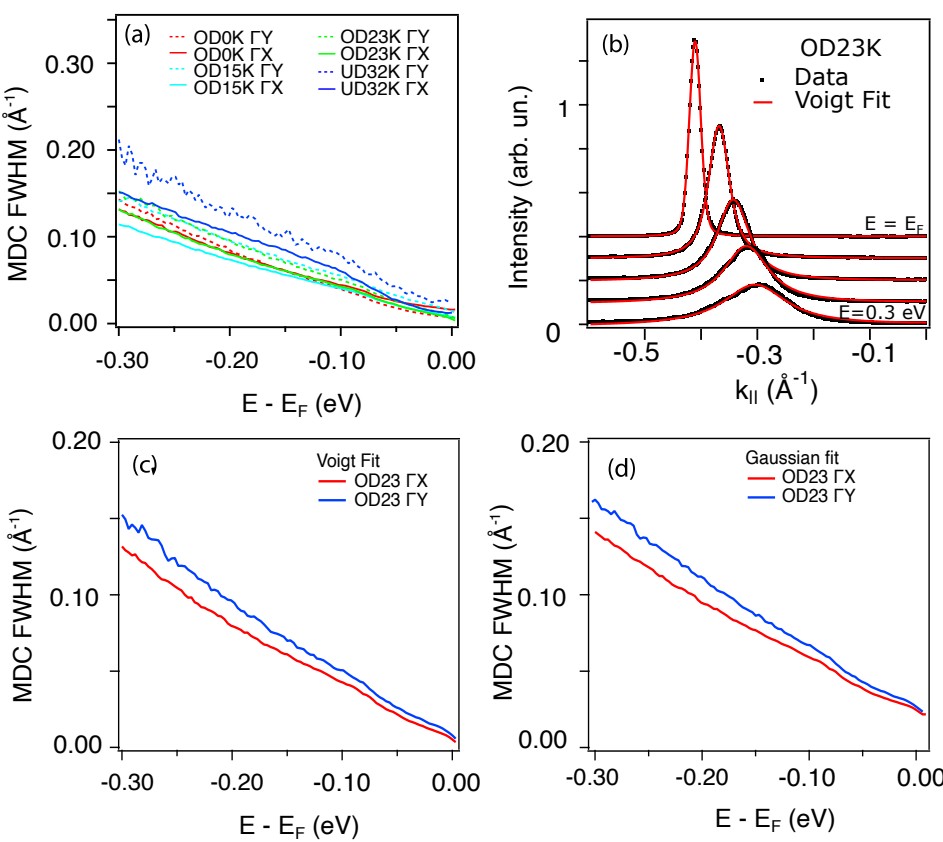

Figure 12: (a) Raw MDC FWHM values for all samples shown in Fig.1(a), without subtracting the $E_F$ values. (b) MDCs from the OD23K sample at five different energies, including fits to a Voigt function. (c) and (d): Comparison of the FWHM between Voigt (c) and Gaussian (d) fits to the data from the OD23K sample in both ΓX and ΓY directions.

Fig. 12 shows the raw MDC FWHM values as a function of energy of all the samples shown in Fig. 1, without any subtraction of MDC width for E=$E_F$. The sample to sample variation of the $E_F$ MDC visible in Fig. 12(a) is attributed to small differences in disorder and/or variation in microscopic sample flatness in the region under the photon beam. In Fig. 12(b) we show a close up view of five exemplary MDC's including Voigt fits for the OD23K sample at energies varying from 0 to 300 meV binding energy. Small deviations of the data (black symbols) to the Voigt functional form (red lines) at high binding energies and momenta are observed, which

originate from the momentum dependence of the self-energy discussed in detail in Ref. [44]. Panel (c) and (d) of Fig. 12 show the extracted absolute value of the FWHM using either the Voigt fits used for the analysis in the main part of the paper or Gaussian fits. The small observed difference between these two fit functions does not influence our conclusions as regards asymmetry in the scattering rates in the two nodal directions.

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
