# Peer review of "The asymmetric Fermi surface of Bi2201"

_SciPost Physics, doi:SciPost Phys. 18, 191 (2025)_

## Round 1 · Referee Report · Anonymous (Referee 1) · 2025-1-28

Report

The manuscript entitled “The asymmetric Fermi surface of (Pby,Bi1−y)2Sr2−xLaxCuO6+δ” investigates the asymmetric Fermi surface of the cuprate Bi2201 using angle-resolved photoemission spectroscopy (ARPES). It reveals a significant asymmetry in Fermi momenta along orthogonal nodal directions, exceeding the orthorhombic distortion observed via X-ray crystallography. Using a tight-binding model and first-principle calculations, the authors propose a mechanism rooted in crystal-field splitting. The study presents a new, linear relationship between ARPES-derived and transport-derived doping estimates, providing insights into the electronic structure of high-temperature superconductors. This manuscript is scientifically rigorous, novel, and well-aligned with SciPost Physics's standards for high-quality research. It makes a significant contribution to the understanding of electronic structure in high-temperature superconductors and fulfills the journal’s publication criteria. With that said, I have some points the authors should clarify: 1. Precision of Manipulator: To perform the cuts along ΓX and ΓY directions, it is necessary to rotate the sample and ensure that the cut is done precisely along the nodal direction. In the method section, it is mentioned that the experiments were conducted at I05 at the Diamond Light Source. How precise is the manipulator? It is worth mentioning this to ensure the robustness of the data. 2. MDC Analysis Presentation: In the paper, only the resultant fit of the MDCs is shown. However, the MDCs and the fitting are not displayed. The authors subtracted the full width at half maximum (FWHM) at EF from the energy-dependent Lorentzian width of the Voigt function fitted to the MDCs. As I understood, this subtraction technique was a key methodological step to ensure the anisotropy measured was intrinsic rather than influenced by extrinsic effects like experimental resolution or sample quality. Can the authors show the MDCs before and after subtraction as a function of energy and temperature, and conduct the same analysis to see if the results obtained are similar? Showing such comparative analysis will strengthen the paper and convince the readers about the robustness of their results. Such comparative analysis can be added as supplementary information. 3. Significance of Linear Relationship: The manuscript introduces a new linear relationship between ARPES-derived and transport-derived doping estimates. Could the authors elaborate on its significance and limitations more explicitly? 4. Quantification of Uncertainties: Systematic uncertainties in ARPES measurements are addressed, but an explicit quantification of experimental and theoretical errors would strengthen the conclusions.

  1. Applicability to Other Cuprates: The paper focuses on Bi2201 and does not fully explore the potential applicability of the findings to other single layer cuprates. If possible, it would be interesting to put these findings in perspective with other cuprates.
  2. Nematicity and Charge Order Discussion: The discussion of nematicity and charge order as alternative explanations for the asymmetry could be expanded to connect with ongoing debates in condensed matter physics. For example, resonant inelastic X-ray scattering (RIXS) studies on Bi2201 have provided significant insights into its electronic properties, particularly concerning charge order and spin excitations. A notable study by Y.Y. Peng et al. (https://doi.org/10.1103/PhysRevB.94.184511) utilized Cu L₃-edge RIXS to investigate charge density modulations in underdoped and optimally doped Bi2201. They observed short-range charge order with a momentum transfer of approximately 0.23 reciprocal lattice units, persisting up to optimal doping levels. This charge order was found to modulate along the Cu-O bond directions, with no evidence of modulation along the nodal (diagonal) direction. Additionally, the out-of-plane measurements indicated a lack of phase correlation, suggesting that the charge order in single-layer Bi2201 is primarily two-dimensional. In the context of this manuscript, the authors reveal a 6-10% difference in the nodal vectors along the ΓY and ΓX directions. This asymmetry was attributed to crystal-field splitting and orthorhombic distortions in the CuO₂ plane. Connecting these findings, the RIXS-detected charge order modulating along the Cu-O bond directions aligns with the ARPES-observed Fermi surface asymmetry. Both studies highlight the significance of anisotropic electronic properties in Bi2201. The charge order observed in RIXS could influence the electronic structure probed by ARPES, potentially contributing to the detected Fermi surface asymmetries. Furthermore, the absence of charge modulation along the nodal direction, as reported in the RIXS study, complements the ARPES findings by indicating that the observed anisotropies are more pronounced along specific crystallographic directions.
  3. In the sentence starting by Berben et al., a subscript is missing for R(T).

Attachment

Recommendation

Ask for minor revision

---

## Round 1 · Referee Report · Anonymous (Referee 2) · 2025-1-29

Report

Smit and co-authors present ARPES measurements on the cuprate superconductor Bi2201. Their data provide evidence for anisotropic electronic structure between the “nodal” directions G-X and G-Y. This shows that the electrons couple to the orthorhombic structural distortion, and should not be treated in a tetragonal scheme. This has direct consequences for evaluation of the carrier density determined by ARPES, which is discussed at length. The anisotropy is also reflected in the self-energy, and persists across a wide range of dopings and temperatures, which suggests that the anisotropy is not related to a phase transition such as nematicity or charge order.

The manuscript is remarkably well-written in a scholarly style, with highly systematic results and analysis. The data is high quality and compellingly evidences the anisotropic electronic structure. The extensive discussion carefully weighs the implications of the results, especially in relation to the well-known Presland relation. Therefore, I highly recommend this manuscript for publication, after my questions are addressed.

  1. The authors should carefully describe the requirements to rule out anisotropy as an experimental artifact. For example, whether it can be derived from anisotropic stray fields in the vacuum chamber, or a distortion in the lens system or detector of the analyzer.
  2. Similarly, the authors should comment on the possible role of photon momentum. The momentum of a 100eV photon is 0.05/Ang, which is the magnitude of the observed effect. Even for the lower photon energies of 28eV employed in Fig 1, the momentum of 0.01/Ang is not insignificant on the scales being discussed.
  3. The authors should show the MDC fitting. In addition, they analyze the Lorentzian component of a Voigt fit. How would the results change with a different curve fit?
  4. Is the carrier density fixed in the comparison of Fig 3? It shows that k_F for the tetragonal structure is smaller than that of both the GX and GY directions of the orthorhombic structure.
  5. The locus of the Fermi surface is determined by MDC peaks, but this may have some error, especially in the antinodal directions where the band dispersion is flatter. Can the authors quantify this error? Perhaps to make it more compelling, the authors could overlay a tight-binding FS with carrier density taken from the Presland relation, and show that it systematically misses the FS.
  6. Since the chemical substitution is non-stochiometric, it necessarily comes with some disorder. How would this manifest in the ARPES spectra and the Luttinger count?
  7. Fig 2d shows a slight doping dependence to the asymmetry. Can this be accounted for simply by shifting the chemical potential in the DFT calculation? It is unclear from the tight-binding analysis since different hopping parameters are used for each doping.

Recommendation

Ask for minor revision

---

## Round 2 · Author Response

Dear Editor,
We thank the referees for providing careful attention to the manuscript, for their many helpful comments and strong recommendation for publication in SciPost Physics. We have made minor revisions and address the referee’s comments below.
Referee 1.
- Precision of Manipulator: To perform the cuts along ΓX and ΓY directions, it is necessary to rotate the sample and ensure that the cut is done precisely along the nodal direction. In the method section, it is mentioned that the experiments were conducted at I05 at the Diamond Light Source. How precise is the manipulator? It is worth mentioning this to ensure the robustness of the data.
We have added a clarifying statement in the text regarding the precision of the manipulator, including a reference to [Hoesch et al, Rev. Sci. Instrum. 88(1) (2017)] where the experimental endstation, including the manipulator, is described in detail.
- MDC Analysis Presentation: In the paper, only the resultant fit of the MDCs is shown.However, the MDCs and the fitting are not displayed. The authors subtracted the full width at half maximum (FWHM) at EF from the energy-dependent Lorentzian width of the Voigt function fitted to the MDCs. As I understood, this subtraction technique was a key methodological step to ensure the anisotropy measured was intrinsic rather than influenced by extrinsic efects like experimental resolution or sample quality. Can the authors show the MDCs before and after subtraction as a function of energy and temperature, and conduct the same analysis to see if the results obtained are similar? Showing such comparative analysis will strengthen the paper and convince the readers about the robustness of their results. Such comparative analysis can be added as supplementary information.
We agree with the referee that the method for MDC fitting is a crucial part of our data analysis and should be made as clear as possible. We would like to point out that the curves shown in Fig.1c are the actual measured MDCs and not fits, and we believe the confusion here is a testament to the quality of our data. To avoid any further confusion, we have now also added the Lorentzian fits to this plot and, as suggested by both of the referees, with a series of energy dependent MDC + fits in supplementary Figure 12. In this Figure, we also now show the FWHM without subtracting the E = Ef value. The sample to sample variation of the widths observed at Ef is attributed to small differences in impurity scattering, i.e. the level of disorder present in each sample and/or variation in microscopic sample flatness in the region under the photon beam. These do not influence our results, but showing them helps to make our analysis method more clear, so we appreciate the suggestion of the referees.
- Significance of Linear Relationship: The manuscript introduces a new linear relationship between ARPES-derived and transport-derived doping estimates. Could the authors elaborate on its significance and limitations more explicitly?
We have added text to the first paragraph of the discussion (page 14), going into further depth on the meaning and significance of the linear off-set in doping. We make a comparison with Tl2201, where a similar extended superconducting dome is measured. The supposed origin for the extended dome for Tl2201, was thought to lie in its inherently cleaner nature, pushing the onset of pair breaking to higher doping. We now suggest this origin to be ruled out for Bi2201 since it is much more disordered. This is also relevant for point 5 of the referee below.
- Quantification of Uncertainties: Systematic uncertainties in ARPES measurements are addressed, but an explicit quantification of experimental and theoretical errors would strengthen the conclusions.
We now explicitly quantify the error caused by the uncertainty in determining the peak-loci at the antinodal locations. Ambiguity of fitting at the anti-node (where the FS is much less defined and difficult to model a contour for) has been used to accurately quantify the uncertainty in the Luttinger count from ARPES, which is shown to be ~±0.025. Figure 8 has been updated with panels (e-f) to show the effect of changing the t2 hopping parameter and its effect on modelling the Fermi surface. This is reflected in the error in pL in figure 5 (b). Added a sentence on page 12 discussing this effect: “A standard error is associated with the calculated pL (±0.025) which arises from the uncertainty in determining the FS at the anti-node. The origin of this uncertainty is illustrated in Fig 8 (e-f).”
- Applicability to Other Cuprates: The paper focuses on Bi2201 and does not fully explore the potential applicability of the findings to other single layer cuprates. If possible, it would be interesting to put these findings in perspective with other cuprates.
We have added to the discussion a section where we compare the shifted dome to the one found in Tl2201, where the extended dome was thought to originate in its inherently cleaner nature, leading to the onset of pair breaking being pushed to higher doping. See also our answer to the referees point 4.
- Nematicity and Charge Order Discussion: The discussion of nematicity and charge order as alternative explanations for the asymmetry could be expanded to connect with ongoing debates in condensed matter physics. For example, resonant inelastic X-ray scattering (RIXS) studies on Bi2201 have provided significant insights into its electronic properties, particularly concerning charge order and spin excitations. A notable study by Y.Y. Peng et al. (https://doi.org/10.1103/PhysRevB.94.184511) utilized Cu L₃-edge RIXS to investigate charge density modulations in underdoped and optimally doped Bi2201. They observed short-range charge order with a momentum transfer of approximately 0.23 reciprocal lattice units, persisting up to optimal doping levels. This charge order was found to modulate along the Cu-O bond directions, with no evidence of modulation along the nodal (diagonal) direction. Additionally, the out-of-plane measurements indicated a lack of phase correlation, suggesting that the charge order in single-layer Bi2201 is primarily two-dimensional. In the context of this manuscript, the authors reveal a 6-10% diference in the nodal vectors along the ΓY and ΓX directions. This asymmetry was attributed to crystal-field splitting and orthorhombic distortions in the CuO₂ plane. Connecting these findings, the RIXS-detected charge order modulating along the Cu-O bond directions aligns with the ARPES-observed Fermi surface asymmetry. Both studies highlight the significance of anisotropic electronic properties in Bi2201. The charge order observed in RIXS could influence the electronic structure probed by ARPES, potentially contributing to the detected Fermi surface asymmetries. Furthermore, the absence of charge modulation along the nodal direction, as reported in the RIXS study, complements the ARPES findings by indicating that the observed anisotropies are more pronounced along specific crystallographic directions.
We agree with the referee: certain translational symmetry breaking effects (such as charge order or nematicity) can in principle distort the Fermi surface. We now explicitly address this case in the manuscript and rule out Charge Order as a cause of the nodal asymmetry we observe. We add the following text to address this: “In the Bi based cuprates specifically, charge density modulations have been observed with resonant inelastic x-ray scattering (RIXS) to persist in underdoped and optimally doped samples, only in the Cu-O bond direction without any out-of-plane component \cite{Peng2016}. Such a translational symmetry breaking can potentially distort the FS, but will be unable to create the type of anisotropy between the two nodal direction we observe as the bandfolding only occurs along the charge order wavevector, and will thus affect the nodal points in the GX~ and GY~ quadrants equally. Together with our observation of clear anisotropy in samples with doping levels of up to p ~ 0.27 rules out these specific ordering phenomena as a root cause. "
Y. Y. Peng, M. Salluzzo, X. Sun, A. Ponti, D. Betto, A. M. Ferretti, F. Fumagalli, K. Kummer, M. Le Tacon, X. J. Zhou, N. B. Brookes, L. Braicovich et al., Direct observation of charge order in under-doped and optimally doped Bi2(Sr, La)2CuO6+δ by resonant inelastic x-ray scattering, Phys. Rev.B 94(18), 1 (2016), doi:10.1103/PhysRevB.94.184511, 1610.01823
- In the sentence starting by Berben et al., a subscript is missing for R(T).
Subscript for $R_{Square}(T)$ is the symbol defined for sheet resistance. This has now been clarified and defined in the text.
Referee 2.
- The authors should carefully describe the requirements to rule out anisotropy as an experimental artifact. For example, whether it can be derived from anisotropic stray fields in the vacuum chamber, or a distortion in the lens system or detector of the analyzer.
We have now more elaborately addressed the precautions taken during the measurements to rule out external experimental artifacts as the source of the observed asymmetry. We have now added text that explicitly addresses this: “Multiple measurements of the nodal 2kf in the two directions have been performed in immediate succession by changing only the azimuthal orientation of the sample, while keeping the exact same beamline and analyser settings. This eliminates extrinsic experimental artifacts, such as stray fields and the influence of the absorbed photon momentum on the measured dispersions, as the source of the observed asymmetry."
- Similarly, the authors should comment on the possible role of photon momentum. The momentum of a 100eV photon is 0.05/Ang, which is the magnitude of the observed effect. Even for the lower photon energies of 28eV employed in Fig 1, the momentum of 0.01/Ang is not insignificant on the scales being discussed.
We have partially addressed this point above for the case of the measured asymmetry. More generally, all ARPES review papers are clear that the photon momentum can safely be neglected for hnu < 100eV, and the transition between the initial and final states is vertical. For soft X-ray energies, k$_{h\nu}$ can make itself felt as a rigid shift in in-plane momenta away from the G point (for example see Fig. 3 in https://www.nature.com/articles/nmat4875). Even if one were to consider that a tiny shift analogous to those visible for hn~500eV and above were to be hidden in our data, this could not yield the observed anisotropy between the GX and GY quadrants we report. Otherwise, ARPES at elevated photon energies (>300eV), and particularly for hard X-ray ARPES (https://www.nature.com/articles/s42005-019-0208-7.pdf) intensity variations due to photoelectron diffraction can be present. These can complicate interpretation of complex, fragmented Fermi surfaces - such as those of Mo metal in the paper just cited. We note that the Bi2201 system we report data on has a very simple band structure with a single band and a single FS, so even were such PED effects to be present - which we stress is not the case at such low photoelectron kinetic energies - they would not present any additional challenge in the data analysis.
- The authors should show the MDC fitting. In addition, they analyze the Lorentzian component of a Voigt fit. How would the results change with a different curve fit
We understand with the point raised by the referees here. We have added the Lorentzian fits to the data of Fig. 1.C, and have added an additional supplemental Figure (Fig.12) explicitly showing our energy dependent MDC fitting. We also show the effect of our choice of fitfunction, by comparing directly the FWHM resulting from the Voigt fits to those from a simple Gaussian profile. Either choice does not affect our conclusions.
- Is the carrier density fixed in the comparison of Fig 3? It shows that k_F for the tetragonal structure is smaller than that of both the GX and GY directions of the orthorhombic structure
We have checked by direct integration of the density of states that according to our calculations the number of electrons per formula unit is the same in both the orthorhombic and the tetragonal structures. This indicates that the characteristic observed by the Referee (namely, the tetragonal Fermi momentum being smaller in the tetragonal case than both the GX and GY directions of the orthorhombic structure) is compensated by changes of the electronic structure in other regions of the Brillouin zone. In the DFT calculations, we do observe changes of the same order of magnitude in the small electron pockets present in the corners of the Brillouin zone.
- The locus of the Fermi surface is determined by MDC peaks, but this may have some error, especially in the antinodal directions where the band dispersion is flatter. Can the authors quantify this error? Perhaps to make it more compelling, the authors could overlay a tight-binding FS with carrier density taken from the Presland relation, and show that it systematically misses the FS.
Figures 4 and 8 have been changed so that the tetragonal Fermi surface (in gold) is not of an arbitrary pL, but in fact show the contour for pLuttinger = pPresland. This shows effectively, as the reviewer suggests, that the Luttinger count derived from Presland significantly deviates from the true measured Fermi surface. The following text is added to page 12 to describe this: “a tetragonal TB model (the same type used for previous studies on Bi-based HTSCs where GX and GY symmetry is assumed) is overlain in gold. The µ in this TB model has been scaled such that pL = pP is shown, thereby clearly illustrating two points: 1) if the hole-doping from the Luttinger count matches the Presland count, the TB model systematically misses the FS entirely, and 2) Visual inspection of the ARPES FS maps confirms the need to incorporate the structural anisotropy and thus that the asymmetry between the electronic states in the Γ X and Γ Y quadrants to be taken into accounting the TB modelling”
- Since the chemical substitution is non-stochiometric, it necessarily comes with some disorder. How would this manifest in the ARPES spectra and the Luttinger count?
Disorder is indeed present in the Bi2201 system, and this mainly manifests itself in the broadening of spectra. Since the widths of the MDC’s at the Fermi level are a measure of the disorder in each sample, this will automatically impose a limit to the accuracy with which we can determine the Luttinger volume and thus the doping. The random scatter of the FWHM at E=Ef, shown in our newly added Fig. 12 is attributed to a combination of disorder and and/or variation in microscopic sample flatness in the region under the photon beam. We do not see a systematic variation as a function of doping for the disorder levels measured.
- Fig 2d shows a slight doping dependence to the asymmetry. Can this be accounted for simply by shifting the chemical potential in the DFT calculation? It is unclear from the tight-binding analysis since different hopping parameters are used for each doping.
We have added the following text to address the apparent doping dependence of the asymmetry: ‘We also note here that the DFT-calculated nodal dispersions in both \GX~ and \GY~ directions (shown in Fig. 3) are approximately linear over a large range in energy (±0.5 eV). This indicates that doping-induced changes of the chemical potential in a rigid band manner cannot explain the small variations of the asymmetry parameter we observe in Fig. 2(d), where the two extremes of our doping series (UD32K and OD0K) seem to have slightly smaller asymmetry parameters compared to the intermediate dopings. At this time, the origin of these small changes is thus an open question.’

---

## Round 2 · List of Changes

[30] M. Hoesch, et al., A facility for the analysis of the electronic structures of solids and their surfaces by synchrotron radiation photoelectron spectroscopy, Rev. Sci. Instrum. 88(1) (2017).
• Page 5: Added fits to the MDCs shown in Fig.1d
• Page 7: Added a discussion regarding the role of charge order in the bismuth based cuprates, including a new reference:
[38] Y. Y. Peng, et al., Direct observation of charge order in underdoped and optimally doped Bi2(Sr, La)2CuO6+δ by resonant inelastic x-ray scattering, Phys. Rev. B 94(18), 1 (2016)
• Page 9: Added a sentence discussing the doping dependence of the Asymmetry parameter. Edited the tetragonal tight-binding model (along with descriptive text) to represent the model presented in figures 4, 8, and table 5.
• Page 11-12: Added a tetragonal TB fit which has PLuttinger = PPresland to the FS of Fig. 4 (including description in caption), as well as a discussion about the significant deviation of the Presland count TB model from the data (i.e. the Fermi surface MDC peaks).
• Page 12-13: Included an in-depth discussion about the source of errors and uncertainties in determining the Luttinger count. An error bar in the x-axis (pPresland)to the OD0K data point in Figure 5 (which was already present in the table for the orthorhombic tight-binding parameters, and is now visualized in the graph). The “error” in the y-axis has been adjusted slightly (pLuttinger uncertainty) to reflect the discussion and the ambiguity of fitting to the anti-node that has been expanded upon in figure 8 (e-f).
• Page 14-15:
Subscript for $R_{Square}(T)$ is the symbol defined for sheet resistance. This has now been clarified and defined in the text.
Added a discussion putting our results in context of other cuprates, specifically a comparison to the shifted dome found in the Tl-based cuprates.
• Page 18: Added citation [44] to the end of section 5.1 as requested by EdAdmin
• Page 20-21 (caption of Figure 6): added citation [82] to section 6.1 as requested by EdAdmin
• Page 23: Added a tetragonal table and updated caption for orthorhombic table (parameters for TB models respectively)
• Page 24: Added tetragonal Presland count TB fits to the FS maps (B, d). Elaborated on uncertainty in TB model fitting by illustrating two different methods of fitting to the anti-node (discussed in depth in pages 12+13). Figure caption updated to reflect these changes.
• Page 27-28: Added section 6.2 and Fig 12. We show the raw FWHM values from the MDC fitting procedure described in the main text, and show exemplary MDC fits as a function of energy. We also compare our Voigt fitting results to those obtained using Gaussian line shapes.

---

## Editorial Decision

published